# Calmodulin regulates TRPV5 intracellular trafficking and plasma membrane abundance

Malou Zuidscherwoude [ID], Teodora Grigore, Brenda van de Langenberg, Guusje Witte, Jenny van der Wijst and Joost G. Hoenderop [ID]

*Department of Medical Biosciences, Research Institute for Medical Innovation, Radboud University Medical Center, Nijmegen, The Netherlands*

Handling Editors: Peying Fong & Pawel Ferdek

The peer review history is available in the Supporting Information section of this article (https://doi.org/10.1113/JP286182#support-information-section).

**Abstract figure legend** Calmodulin regulates TRPV5 intracellular trafficking and plasma membrane abundance. TRPV5 channels traffic through the secretory and endocytic recycling pathway, although they are predominantly located in the endoplasmic reticulum (ER). Upon CaM overexpression, TRPV5 is retained in the ER. In accordance, upon CaM knockdown, or disruption of TRPV5–CaM interaction, TRPV5 abundance on the plasma membrane is increased. This study reveals a novel role for CaM in $Ca^{2+}$-dependent TRPV5 regulation, modulating TRPV5 intracellular trafficking. Created in BioRender.com.

**Abstract** As a member of the transient receptor potential (TRP) superfamily of ion channels, TRPV5 is a unique $Ca^{2+}$-selective channel important for active reabsorption of $Ca^{2+}$ in the kidney. TRPV5-mediated $Ca^{2+}$ entry into the cell is controlled by a negative feedback mechanism, in which calmodulin (CaM) blocks the TRPV5 pore upon $Ca^{2+}$ binding. Combining microscopy techniques and biochemical assays, the present study uncovered an auxiliary role for CaM in the regulation of human (h)TRPV5 intracellular trafficking. Overexpressed hTRPV5 was mainly localised to the endoplasmic reticulum (ER) and associated with peripheral ER tubules. Limiting expression using the HEK293 TET-off system revealed that hTRPV5 trafficked through the endocytic recycling pathway. CaM co-localised with hTRPV5 at intracellular sites and overexpression of CaM slowed hTRPV5 exit from the ER. In accordance, CaM binding-disrupting truncations of the TRPV5 C-terminus (698X) or knockdown of endogenous CaM by small interfering RNA resulted in an increased

fraction of TRPV5 that localised to the plasma membrane. hTRPV5 expressing cells had an increased intracellular $Ca^{2+}$ concentration upon knockdown of CaM. The protein abundance of the $Ca^{2+}$ impermeable hTRPV5-D542 mutant is also regulated by CaM, which suggests that the mode of action is independent of disrupted intracellular calcium concentrations. In conclusion, our study reveals a novel role for CaM in $Ca^{2+}$-dependent TRPV5 regulation, modulating TRPV5 intracellular trafficking.

(Received 22 December 2023; accepted after revision 29 October 2024; first published online 21 November 2024)
**Corresponding author** J. G. Hoenderop: Department of Medical Biosciences, Research Institute for Medical Innovation, Radboud University Medical Centre, PO Box 9101, 6500 HB, Nijmegen, The Netherlands.    Email: joost.hoenderop@radboudumc.nl

## Key points

- The renal $Ca^{2+}$ channel TRPV5 is a crucial player in maintenance of the body's $Ca^{2+}$ homeostasis.
- $Ca^{2+}$ transport through TRPV5 is controlled by single channel activity, as well as TRPV5 plasma membrane abundance.
- Calmodulin (CaM) co-localised with TRPV5 at intracellular sites and retained TRPV5 in the endoplasmic reticulum.
- Disrupted CaM–TRPV5 binding or knockdown of endogenous CaM by small interfering RNA (siRNA) resulted in an increased TRPV5 plasma membrane abundance.
- Knockdown of endogenous CaM by siRNA resulted in increased intracellular $Ca^{2+}$ concentrations. The regulation of TRPV5 trafficking by CaM is independent of the effect of CaM on intracellular $Ca^{2+}$ concentrations.
- This study reveals a novel role for CaM in $Ca^{2+}$-dependent TRPV5 regulation, next to its ability to directly block the TRPV5 channel pore, by modulating TRPV5 trafficking in the secretory pathway.

## Introduction

As a member of the transient receptor potential (TRP) family, TRPV5 stands out as one of the most $Ca^{2+}$-selective of these ion channels and plays a crucial role in $Ca^{2+}$ homeostasis as the gatekeeper for $Ca^{2+}$ reabsorption in the kidney. TRPV5 facilitates $Ca^{2+}$ entry at the apical membrane of cells of the distal convoluted tubule (DCT) and connecting tubule (CNT), the sections of the nephron where the final urinary $Ca^{2+}$ excretion is fine-tuned (Downie & Alexander, 2022; Hoenderop et al., 1999). The importance of TRPV5 is evidenced by *Trpv5* knockout mice, which present with severe hypercalciuria, together with bone abnormalities, significantly higher levels of 1,25-dihydroxyvitamin $D_3$ and compensatory intestinal $Ca^{2+}$ hyperabsorption (Hoenderop et al., 2003; Renkema et al., 2005). Furthermore, polymorphisms in the human *TRPV5* gene are associated with the risk of developing kidney stones (Ali et al., 2022; Mandal et al., 2022; Mitra et al., 2020; van der Wijst et al., 2019).

TRPV5 contains classic TRPV protein family features including six transmembrane domains, amino (N)-terminal ankyrin repeats and the TRP domain (Hughes, Lodowski et al., 2018), as well as the need to tetramerise to form a functional channel. TRPV5 and its close homolog TRPV6 are distinct from other members of the TRPV family because of their high selectivity for $Ca^{2+}$. Moreover, in contrast to TRPV1–4, TRPV5 and TRPV6 do not exhibit thermosensitivity or ligand-dependent activation but show constitutive activity at physiological membrane potentials. This open conformation is maintained by the interaction with its endogenous co-factor, the membrane phospholipid phosphatidylinositol 4,5-bisphosphate $(PI(4,5)P_2)$ (Cai et al., 2020; Hughes, Pumroy et al., 2018; Rohacs, 2023).

Upon $Ca^{2+}$ entry, TRPV5 undergoes rapid inactivation *via* a $Ca^{2+}$-dependent feedback mechanism, which is mediated by the interaction with $Ca^{2+}$ binding protein calmodulin (CaM) (Lambers et al., 2004; Nilius, Prenen, et al., 2001). CaM consists of two lobes connected with a flexible linker, with each lobe containing two EF hand domains that are capable of binding $Ca^{2+}$. CaM transduces signals to its target proteins in accordance with cytosolic $Ca^{2+}$ levels by adopting conformational changes upon $Ca^{2+}$ binding (Hoeflich & Ikura, 2002). In the current model of TRPV5 inhibition, CaM can

pre-associate with TRPV5 in a $Ca^{2+}$-free state (apoCaM). When local intracellular $Ca^{2+}$ concentrations rise, the carboxyl (C)-lobe of CaM is calcified first, which increases the binding of CaM to the C-terminus of TRPV5. The concerted action of both CaM lobes leads to a conformational change in the C-terminus of TRPV5, leading to obstruction of the TRPV5 pore by the side chain of K115 of the CaM C-lobe (Dang et al., 2019; Hughes, Pumroy et al., 2018; Zuidscherwoude et al., 2023). This inactivation of TRPV5 channels can be regulated by modulating the interaction between TRPV5 and CaM. Parathyroid hormone stimulates active $Ca^{2+}$ reabsorption via the adenylyl cyclase–cAMP–protein kinase A pathway, leading to phosphorylation of TRPV5 at threonine-709 which prevents CaM binding (de Groot et al., 2009).

TRPV5 is strongly regulated at the level of single channel activity, as well as by controlled membrane abundance. The vast majority of TRPV5 channels are stored in intracellular membranes and presence on the cell surface is dynamically regulated by endosomal recycling processes (Lambers et al., 2007; van de Graaf et al., 2008). Previous studies uncovered diverse molecular mechanisms for calciotropic hormones and other factors to act on TRPV5 membrane abundance, including increasing TRPV5 gene expression, membrane anchoring and inhibition of endocytosis (Al-Bataineh et al., 2023; Andrukhova et al., 2014; Boros et al., 2012; Cha & Huang, 2010; de Groot et al., 2008; Hoover et al., 2016; Jing et al., 2011; Lee et al., 2021; Leunissen et al., 2013, 2016; Nie et al., 2016; Radhakrishnan et al., 2013; Tudpor et al., 2012; Wolf et al., 2013, 2014). However, little is known about TRPV5 trafficking to the plasma membrane directly after channel assembly in the secretory pathway. Because high intracellular $Ca^{2+}$ levels are cytotoxic, we hypothesise that, next to $Ca^{2+}$-dependent inactivation, CaM is involved in regulating the rate of plasma membrane insertion of new TRPV5 channels. This hypothesis is strengthened by studies uncovering a role for CaM in the trafficking of other ion channels including TRPV1, CaV1.2, NaV1.4, KCNQ2, $Ca^{2+}$-activated $K^+$ channels and aquaporins (Biswas et al., 2008; Etxeberria et al., 2008; Lee et al., 2003; Markou et al., 2022; Rosenbaum et al., 2004; Wang et al., 2007). The present study aimed to identify the secretory pathway of human TRPV5 and whether CaM regulates TRPV5 trafficking and membrane abundance.

## Methods

### Molecular cloning

Restriction enzyme digestions, DNA ligations and other recombinant DNA procedures were performed using standard protocols. Human TRPV5 (van Goor et al., 2017) was cloned into a pC1-eGFP vector containing a N-terminal eGFP tag. Subsequently, eGFP-hTRPV5 was cloned in the pTRE3G vector (Takara Bio Europe, Saint-Germain-en-Laye, France). HA-tagged Rabbit TRPV5 and HA-tagged rabbit TRPV5 W702A, R706E, T709A and 698X have been described previously (de Groot et al., 2011; van der Wijst et al., 2017). Rat CaM and the $Ca^{2+}$-insensitive mutant CaM1234, based on mutations in the EF hand loops (D20A, D56A, D93A and D129A) (Geiser et al., 1991), were cloned into a pC1-mCherry vector containing a N-terminal mCherry. mIFP-calnexin-N-14 was a gift from Michael Davidson (Yu et al., 2015) (plasmid #56214; Addgene, Watertown, MA, USA). The D542A mutation was introduced in pC1-eGFP-hTRPV5 with a Q5 site directed mutagenesis kit (New England Biolabs, Frankfurt am Main, Germany) using the following primers, forward: CACTGTTATTGCAGCACCTGCCACC, reverse: AGAAAAAGCTCAAAGGTGG. All DNA constructs were verified by DNA sequencing. DNA for mammalian cell transfection was amplified in *Escherichia coli* TOP10f strain and plasmid preparation was performed using the Nucleobond Xtra Midi kit in accordance with the manufacturer's instructions (Macherey-Nagel, Düren, Germany).

### Buffers

Permeabilisation buffer: 0.3% (v/v) Triton X-100 in PBS. Goat serum dilution buffer (GSDB): 16% (v/v) goat serum and 0.3% (v/v) Triton X-100 in PBS. PBS-CM: 1 mm $MgCl_2$, 0.5 mm $CaCl_2$ in PBS, pH 8.0 adjusted with NaOH. Triton lysis buffer: 50 mm Tris-HCl (pH 7.5), 270 mm sucrose, 150 mm NaCl, 10 mm sodium-glycerophosphate, 1 mm EGTA, 1 mm EDTA, 1% (v/v) Triton X-100, freshly supplemented with phenylmethylsulfonyl fluoride (1 mm), leupeptin (2.34 μm), aprotinin (0.15 μm) and pepstatin-A (1.46 μm). Laemmli sample buffer (5×): 0.3 m Tris-HCl (pH 6.8), 50% (v/v) glycerol, 10% (w/v) SDS, 0.05% (w/v) bromophenol blue and 0.5 m DTT. TBS-Tween (TBS-T): Tris-HCL (200 mm, pH 7.5), 150 mm NaCl and 0.1% (v/v) Tween-20. Hanks' balanced salt solution (HBSS): 8 g $L^{-1}$ NaCl, 400 mg $L^{-1}$ KCl, 350 mg $L^{-1}$ NaHCO$_3$, 60 mg $L^{-1}$ KH$_2$PO$_4$, 48 mg $L^{-1}$ Na$_2$HPO$_4$, 1 g $L^{-1}$ glucose. Fura-2 EDTA buffer: 0.4 mm EDTA in HBSS. Fura-2 $Ca^{2+}$ buffer: 4.2 mm $CaCl_2$ in HBSS.

### Antibodies

Primary antibodies were: mouse anti-PDI (ER, RL90; Thermo Fisher Scientific Inc., Waltham, MA, USA; dilution 1:1000), mouse anti-G58K (Golgi apparatus, G2404; Sigma-Aldrich, St Louis, CA, USA; dilution 1:200), mouse anti-EEA1 (early endosomes, E41120-050; BD Transduction Laboratories, San Diego, USA;

dilution 1:200), rabbit anti-Rab11 (recycling endosomes, D4F5; Cell Signaling Technology, Danvers, MA, USA; dilution 1:50), mouse anti-LAMP1 (lysosomes, H4A3-S; Developmental Studies Hybridoma Bank, Iowa, IA, USA), anti-calmodulin (05-173, Upstate; Merck KGaA, Darmstadt, Germany; dilution 1:500), rabbit anti-eGFP (G1544; Sigma-Aldrich; dilution 1:5000), mouse anti-HA tag (2367; Cell Signaling Technology; dilution 1:5000), mouse anti-ß-actin (A5441; Sigma-Aldrich; dilution 1:2000). Secondary antibodies were: Alexa Fluor 647-conjugated goat-anti-mouse-IgG (A-21236; Invitrogen, Thermo Fisher Scientific Inc.; dilution 1:400), Alexa Fluor 647-conjugated goat-anti-rabbit-IgG (Thermo Fisher Scientific Inc.; dilution 1:400), peroxidase-conjugated goat-anti-rabbit-IgG (A4914; Sigma-Aldrich; dilution 1:10 000), peroxidase-conjugated sheep-anti-mouse-IgG (Jackson ImmunoResearch, Ely, UK; dilution 1:10 000).

### Cell culture

Human embryonic kidney (HEK)293 cells were grown in Dulbecco's modified Eagle's medium (Thermo Fisher Scientific Inc.) enriched with 10% (v/v) fetal bovine serum (Greiner Bio-One, Alphen aan de Rijn, The Netherlands), 2 mM non-essential amino acids (Sigma-Aldrich), 2 mM L-glutamine (Sigma-Aldrich) and 1 mM sodium pyruvate (Life Technologies Limited, Renfrew, UK), at 37°C and 5% (v/v) $CO_2$. Cells were not used after 20 passages. HEK293 TET-off cells (#631152; Takara Bio Europe) were cultured under the same conditions, with 100 μg mL$^{-1}$ G418-sulphate Geneticin (Life Technologies Limited), to maintain the TET-off transactivator.

### Cell plating and transfection

Cells were seeded in six-well plates or on fibronectin (Roche Diagnostics, Rotkreuz, Switzerland) coated 18 mm round coverslips (#1.5; Epredia, Portsmouth, NH, USA) in a 12-well plate (Greiner Bio-One). Cells were transfected using Lipofectamine 2000 (Thermo Fisher Scientific Inc.) in accordance with the manufacturer's instructions with a μg DNA to μL lipofectamine ratio of 1:2.5 for HEK293 cells and 1:2 for HEK293 Tet-Off cells.

**TRPV5 intracellular localisation.** HEK293 cells were seeded in a six-well plate and co-transfected with 0.5 μg of pC1-eGFP-hTRPV5 and 0.5 μg of pC1-mCherry-CaM wild-type (WT) for the confocal experiment, or co-transfected with 1 μg of pC1-eGFP-hTRPV5, 0.5 μg of pC1-mCherry-CaM WT and 1 μg of mIFP-calnexin-N-14 for the Airyscan experiment. After 24 h, the cells were re-plated using accutase (Sigma-Aldrich) on fibronectin coated coverslips and fixated after 6 h.

**CaM and TRPV5 Co-localisation.** HEK293 cells were seeded on fibronectin coated coverslips. After 24 h the cells were co-transfected with 0.5 μg of pC1-eGFP-hTRPV5 and 0.5 μg of pC1-mCherry-Mock, pC1-mCherry-CaM WT or pC1-mCherry-CaM1,2,3,4 and fixated after 24 h.

### Cell fixation and immunocytochemistry

Cells were washed three times in PBS before fixation with 4% paraformaldehyde (PFA) (Sigma-Aldrich) in PBS (10 min at room temperature (RT)). After three washes in PBS, autofluorescence of PFA was quenched by incubating the coverslips with 50 mM $NH_4Cl$ in PBS (10 min at RT). Coverslips were washed three times with PBS and were subjected to immunocytochemistry or were embedded using Fluoromount-G (Thermo Fisher Scientific Inc.). For immunocytochemistry, cells were incubated with permeabilisation buffer (10 min at room temperature), washed three times and blocked using GSDB (30 min at RT). Cells were incubated with primary antibodies for 1 h in GSDB at RT, washed three times in PBS and incubated with secondary antibodies for 45 min in GSDB at RT. After washing with PBS, the cells were embedded with Fluoromount-G.

### Pulse expression

HEK293 Tet-off cells were seeded on fibronectin coated coverslips. After 12 h, cells were transfected with 1.5 μg of pTRE3G-eGFP-hTRPV5 for immunocytochemistry experiments or co-transfected with 1.5 μg of pTRE3G-eGFP-hTRPV5, 750 ng of mIFP-calnexin-N-14 and 500 ng of pC1-mCherry-Mock, pC1-mCherry-CaM WT or pC1-mCherry-CaM1,2,3,4. After 10 h, transcription of the pTRE3G vector was stopped by the addition of 1 μg mL$^{-1}$ doxycycline hyclate (Sigma-Aldrich). Cells were fixated directly, 4 or 8 h after doxycycline addition.

### Confocal and Airyscan microscopy

Confocal microscopy was performed on a Leica-SP8 SMD system (Leica, Wetzlar, Germany) with a 63×/1.2 numerical aperture (NA) Plan Apochromat water objective lens (Carl Zeiss AG, Jena, Germany). Samples were excited at 488, 561 and 635 nm by a high energy pulsed infrared-fibre white laser and emitted light was captured by hybrid detectors. Airyscan imaging was performed on a Zeiss LSM 880 confocal microscope equipped with a 63× Plan Apochromat (1.4 NA) oil objective and a 32-channel GaAsP Airy detector (Carl Zeiss AG).

## Small interfering RNA (siRNA) treatment

HEK293 cells were plated in six-well plates and transfected with a total concentration of 75 nM siRNA using 5 μL of lipofectamine RNAiMAX (Invitrogen) in accordance with the manufacturer's instructions. For CaM knockdown, 25 nM CALM1, 25 nM CALM2 and 25 nM CALM3 siRNA (siGENOME SMARTpool; Dharmacon, Lafayette, CO, USA) was used and, as a control 75 nM Non-Targeting (NT) siRNA (siGENOME Pool #1; Dharmacon) was transfected into HEK293 cells. After 24 h, cells were transfected with 1 μg of pC1-eGFP-hTRPV5 or pC1-eGFP-hTRPV5-D542A for biotinylation assays, or 0.5 μg of pC1-eGFP-hTRPV5 with either 0.5 μg of pC1-mCherry or 1 μg of pN1-mIFP. Knockdown efficiency was determined via western blotting.

## Cell surface biotinylation assay

Cells were subjected to cell surface biotinylation 20 h after TRPV5 transfection. All steps were performed at 4°C with pre-cooled reagents. Cells were washed twice in PBS-CM before incubation with 0.5 mg mL$^{-1}$ sulfo-NHS-LC-LC-biotin (Thermo Fisher Scientific Inc.) in PBS-CM for 30 min. Unbound biotin was removed by washing the cells twice with 0.1% (w/v) bovine serum albumin in PBS-CM and twice with ice-cold PBS (pH 7.3). After this, the cells were lysed using Triton lysis buffer and protein concentrations were determined using a BCA protein assay kit (Thermofisher Scientific Inc.) in accordance with the manufacturer's protocol, and subsequently equalised and an input sample was taken. Equal protein amounts were incubated with neutravidin agarose beads (Thermo Fisher Scientific; 20 μL per sample) for 2 h at 4°C under gentle rotation. Following four washes of the neutravidin agarose beads with lysis buffer, proteins were eluted in 2× Laemmli sample buffer.

## Western blotting

Protein samples were subjected to 10% (w/v) SDS-PAGE and transferred to polyvinylidene fluoride membranes. These membranes were blocked for 45 min with 5% (w/v) non-fat dry milk (NFDM) in TBS-T at RT, and immunoblotted overnight at 4°C with primary antibodies in 5% (w/v) NFDM TBS-T. The membranes were washed with TBS-T, and incubated with peroxidase-labelled secondary antibodies in 5% (w/v) NFDM TBS-T for 1 h at RT. Following washes with TBST-T and TBS, protein expression was visualised by chemiluminescence using SuperSignal West reagent (Thermo Fisher Scientific Inc.)

on an Image Quant 4000 (GE Healthcare Bio-Sciences AB, Uppsala, Sweden).

## Intracellular Ca$^{2+}$ measurements using Fura-2-AM

Six hours after transfection with eGFP-TRPV5, cells were washed and detached using accutase. NT and CaM knockdown cells or TRPV5 WT and TRPV5 D542A cells were mixed 1:1 and plated in fibronectin coated WillCo dishes (WillCo Wells, Amsterdam, The Netherlands). After 16 h, cells were loaded with 3 μM Fura-2-AM (Molecular Probes, Eugene, OR, USA) in cell culture media with 0.01% (v/v) Pluronic F-129 (Molecular Probes) at 37°C for 20 min. After loading, the cells were washed with HBSS and allowed to equilibrate for 10 min in 1 mL of Fura-2 EDTA buffer. Cells were placed on an inverted microscope on a stage incubator at 37°C (Zeiss Axio Observer 7 with Zeiss Axiocam 702) and, prior to live cell imaging, the eGFP, mCherry and mIFP were imaged for cell identification. Next, the Fura-2 probe was excited at 340 and 380 nm by using a Lambda DG5 light source (Sutter, Novato, CA, USA), and visualised at 20× magnification with a sampling interval of 2 s. At the tenth frame, 1 mL of Fura Ca$^{2+}$ buffer was added.

## Statistical analysis

Microscopy data and optical densities of western blot bands were quantified using Fiji Image J, version 2.14.0 (Schindelin et al., 2012). For confocal microscopy data, the plugin JACoP (Dunn et al., 2011) was used for quantification of co-localisation and co-occurrence. For the Manders' coefficient, single cells were analysed and positive pixels for eGFP and mIFP/alexa 647 were selected by manual thresholding. The fraction of eGFP positive pixels that were also positive for mIFP/alexa 647 was determined. For the Pearson's coefficient, single cells were analysed. Cellular areas of interested were automatically selected by Costes Mask. The nucleus was excluded from this region of interest (ROI). The correlation between the pixel intensity of eGFP and mCherry was quantified. For the analysis of intracellular Ca$^{2+}$ measurements, ROI in single eGFP expressing cells were selected and cells were annotated as NT or siRNA treated cells by mCherry or mIFP expression. After background correction, the fluorescence emission ratio of 340 and 380 nm excitation was calculated for every frame as read-out for the intracellular Ca$^{2+}$ concentration. $P < 0.05$ was considered statistically significant. Data representation and statistical analysis were performed in Prism, version 9.4.1 (GraphPad Software Inc., San Diego, CA, USA).

# Results

## The intracellular localisation of human TRPV5

The number of TRPV5 channels on the plasma membrane needs to be carefully regulated to prevent $Ca^{2+}$ cytotoxicity. Accordingly, previous studies using mammalian cell lines transfected with rabbit TRPV5 noticed a predominantly intracellular location of TRPV5 (Lambers et al., 2007; van de Graaf et al., 2006). We applied confocal microscopy to pinpoint the location of human TRPV5 at the subcellular level. To this end, we transfected HEK293 cells with eGFP-tagged hTRPV5 and mCherry-tagged CaM and identified relevant cellular organelles via immunocytochemistry (Fig. 1A) using

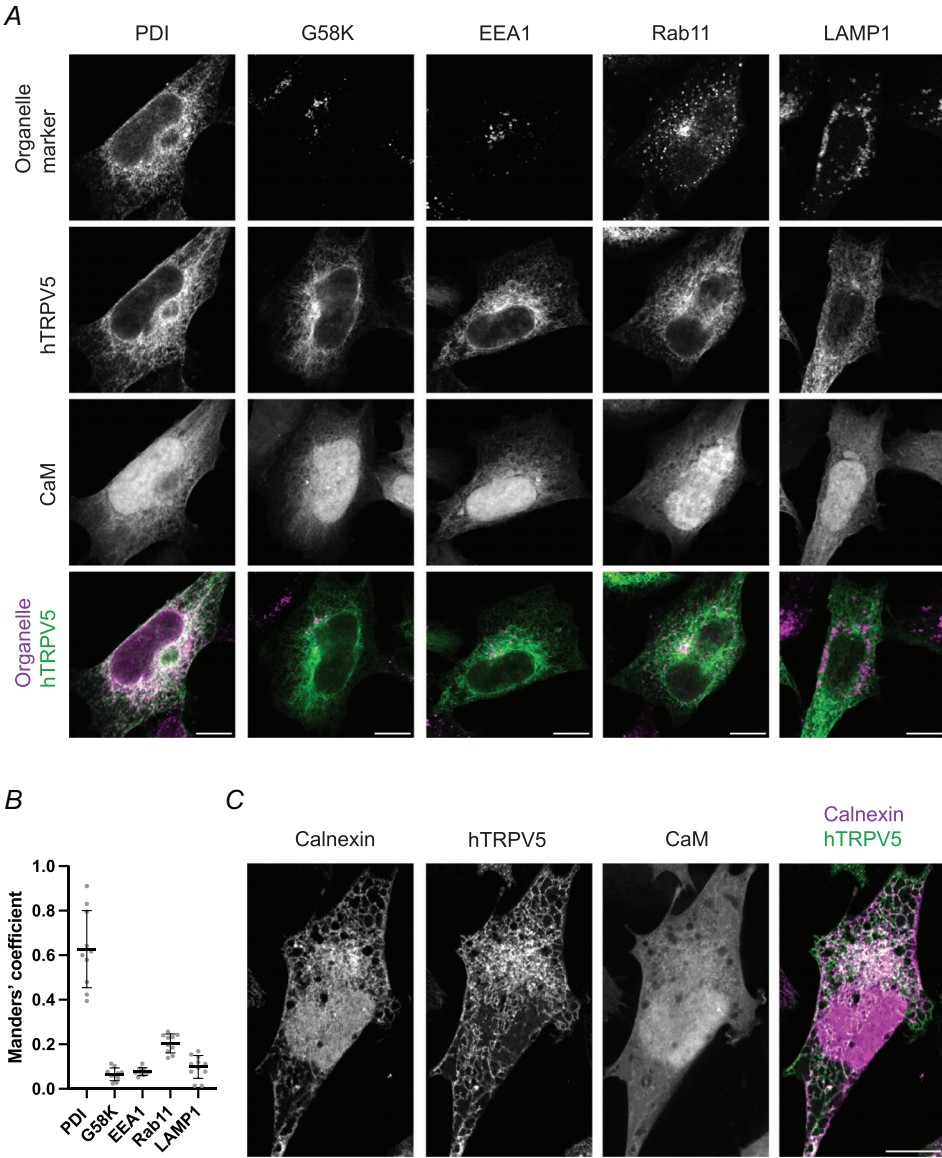

**Figure 1. Association of hTRPV5 with the ER tubular network and the endocytic recycling pathway**
*A*, representative confocal microscopy images of HEK293 cells transfected with eGFP-hTRPV5 and mCherry-CaM with the location of intracellular organelles visualised by immunocytochemistry. ER, PDI; Golgi apparatus, G58K; early endosomes, EEA1; recycling endosomes, Rab11; lysosomes, LAMP1. Bottom row displays merged images of eGFP-hTRPV5 in green with the indicated organelle marker in magenta and co-localisation in white. Scale bar = 10 µm. *B*, fraction of eGFP-hTRPV5 residing in the studied organelles was calculated using Manders' coefficient. Fraction of eGFP positive pixels overlapping with pixels positive for the organelle marker. Number of cells ≥10; mean ± SD. Representative experiment out of three independent experiments. *C*, HEK293 cells transfected with eGFP-hTRPV5, mCherry-CaM and genetic ER marker calnexin-mIFP. Unpermeabilised cells were visualised by Airyscan confocal imaging. Right image displays a merge between the eGFP-hTRPV5 (green) and the calnexin-mIFP (magenta) images. Scale bar = 10 µm. Representative out of eight cells analysed in two independent experiments. [Colour figure can be viewed at wileyonlinelibrary.com]

antibodies against protein disulphide isomerase (PDI, ER), G58K (Golgi), EEA1 (early endosomes), Rab11 (recycling endosomes) and LAMP1 (lysosomes). We co-transfected CaM in this microscopy based study because the robust (over)expression of hTRPV5 needed for intracellular visualisation caused cells to round up, hampering detailed localisation analyses. TRPV5 was found throughout the cytoplasm, with the highest intensity found in cytoplasmic regions close to the nucleus (Fig. 1*A*). Importantly, a plasma membrane localisation was not clearly observable. The co-occurrence of TRPV5 with the organelles was quantified by the Manders' coefficient, which describes the fraction of all the TRPV5 positive pixels also positive for the organelle marker. The larger part of eGFP-TRPV5 was overlapping with the location of the ER marker PDI (Fig. 1*B*). Furthermore, a smaller fraction of TRPV5 was associated with recycling endosomes.

### The association of TRPV5 with the ER tubule network

The intracellular localisation of TRPV5 is remarkable because, alongside vesicles, a TRPV5 positive reticular network is observed in the cytoplasm of the cells. To interrogate the subcellular location of TRPV5 in relation to the ER with an increased resolution, we employed Airyscan confocal microscopy on unpermeabilised cells. HEK293 cells were triple transfected with eGFP-tagged human TRPV5, mCherry-tagged CaM and the ER marker calnexin tagged with mIFP (Yu et al., 2015). An optical section below the nucleus was chosen to observe the entire ER network, consisting of the nuclear envelope, perinuclear ER domain and a peripheral ER tubular network. Co-localisation analysis with calnexin demonstrated that TRPV5 was not localised in the nuclear envelope but was associated with perinuclear and peripheral ER tubules (Fig. 1*C*, right).

### Trafficking of TRPV5 to the endocytic pathway

Overexpression of TRPV5 may skew its intracellular location. By limiting the time allowed for transcription, we were able to reduce the expression of TRPV5 and track the subcellular location of TRPV5 over time. The HEK293 TET-off cell line with the TET-off expression system allowed for doxycycline-mediated transcription inhibition. Upon addition of doxycycline to these cells, the interaction between the transcription factor tetracycline transactivator (TTA) and the TET-responsive element (TRE) promotor, and therefore the transcription of TRE-containing plasmids, is inhibited. Indeed, in HEK293 TET-off cells transfected with pTRE3G-eGFP-hTRPV5, protein expression of eGFP-hTRPV5 was abolished when cells were treated

with doxycycline immediately after plasmid transfection (Fig. 2*A*). To fine-tune the levels of TRPV5 protein expression, doxycycline was added 10 h after transfection. In this setup, eGFP-TRPV5 protein expression was reduced compared to the overexpression observed in untreated cells (Fig. 2*A*).

To study the intracellular trafficking of TRPV5, HEK293 TET-off cells were transfected with pTRE3G-eGFP-hTRPV5 and transcription was stopped after 10 h by the addition of doxycycline. This pulse expression of eGFP-TRPV5 allowed us to track the location of TRPV5 at different timepoints after translation at the same time as preventing the filling up of the ER with newly synthesised TRPV5 protein. Over time, the distribution of TRPV5 within the cell shifted from a perinuclear and tubular network location to a punctate pattern (Fig. 2*B*).

To investigate TRPV5 forward trafficking in more detail, cells were analysed 0, 4 or 8 h after doxycycline addition and subjected to immunocytochemistry to identify co-occurrence of TRPV5 with cellular organelle markers. Ten hours after transfection (time after doxycycline $t = 0$), a large fraction of eGFP-TPRV5 was overlapping with the ER marker PDI (Fig. 2*C*). Scarcely any eGFP-TRPV5 was associated with early endosomes or Rab11 positive recycling endosomes (Fig. 2*D* and *E*). Over the timespan of 8 h, the fraction of TRPV5 overlapping with PDI reduced significantly indicating that TRPV5 trafficked out of the ER ($P = 0.00378$) (Fig. 2*F*). The fraction of TRPV5 residing in EEA1 positive vesicles increased ($P = 0.0374$). At 4 and 8 h after doxycycline addition, a significantly larger fraction of TRPV5 was associated with recycling endosomes, compared to 0 h after doxycycline addition, indicating that TRPV5 rapidly enters the endocytic recycling pathway after translation ($t = 0$ *vs.* $t = 4$, $P = 0.00178$; $t = 0$ *vs.* $t = 8$, $P = 0.00904$) (Fig. 2*F*).

### Subcellular co-localisation of TRPV5 and CaM

Over the last decades, several groups demonstrated the interaction between CaM and rabbit TRPV5, using biochemical studies, cryo-electron microscopy and NMR (de Groot et al., 2011; Rohacs et al., 2022). However, these studies are limited by the lack of cellular context or the use of truncated termini of the TRPV5 channel. Here, confocal microscopy was performed to assess the colocalisation pattern of mCherry-CaM with eGFP-tagged hTRPV5 channels in intact HEK293 cells. CaM had a whole-cell localisation with a large fraction inside the nucleus (Figs 1 and 3*A*). CaM significantly co-localised with hTRPV5, as indicated by a Pearson's coefficient different to 0 ($P < 0.001$) and different to the control condition: hTRPV5 co-transfected with mCherry

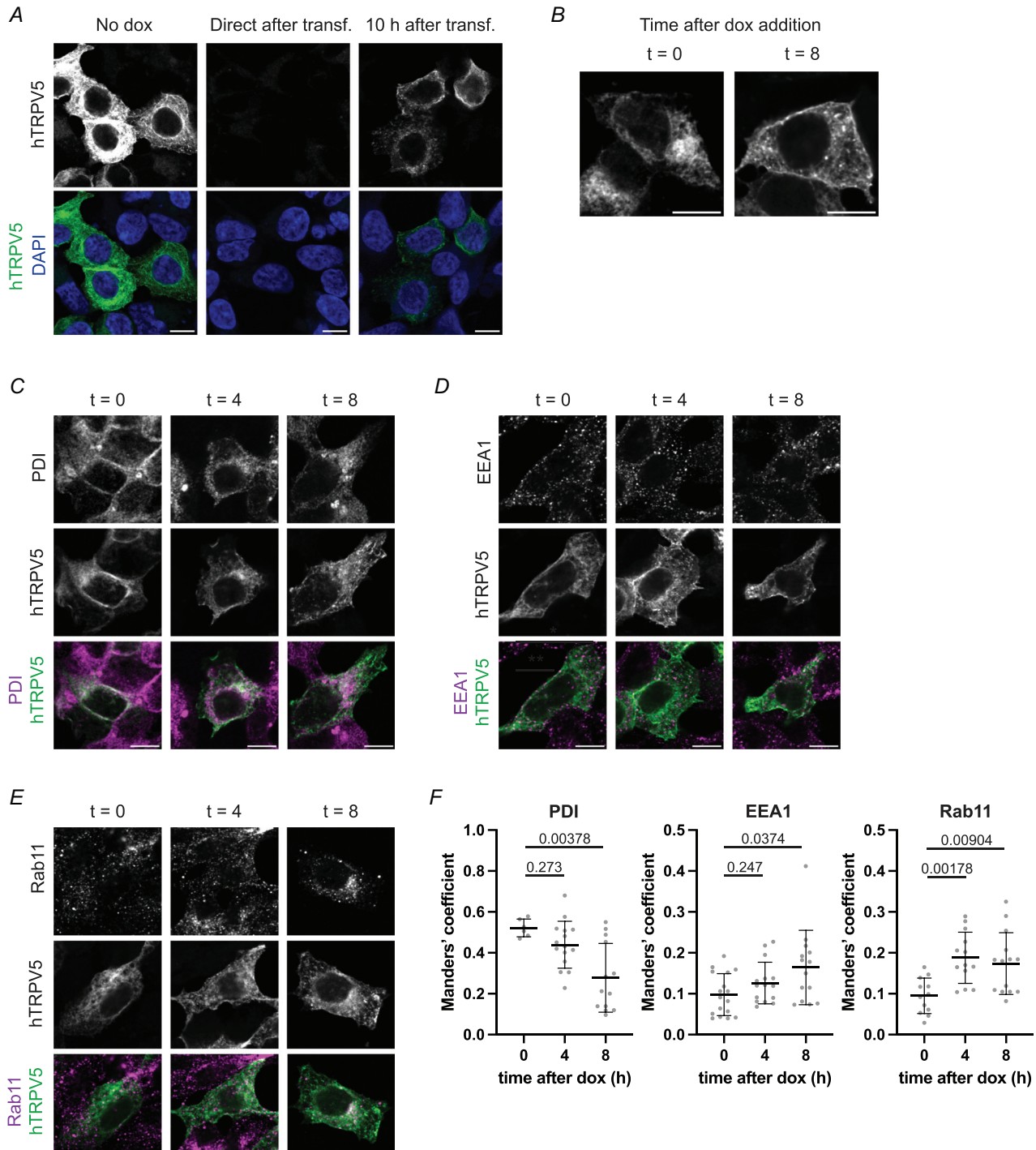

**Figure 2. TRPV5 ER exit and entry of the endocytic pathway after translation**
*A*, representative confocal microscopy images of HEK293 TET-off cells transfected with pTRE3G-eGFP-hTRPV5. Doxycycline was not added or either added directly or 10 h after transfection to stop transcription from the pTRE3G promotor. eGFP-hTRPV5 was visualised 18 h after transfection. Scale bar = 10 μm. *B–F*, HEK293 TET-off cells were transfected with pTRE3G-eGFP-hTRPV5 and transcription was stopped after 10 h by doxycycline addition. Cells were fixed immediately (*t* = 0), 4 or 8 h after doxycycline addition (*t* = 4 and 8) and subjected to immuno-cytochemistry with the organelle markers PDI (*C*), EEA1 (*D*) or Rab11 (*E*). Representative images; the scale bar = 10 μm. *F*, fraction of eGFP-hTRPV5 residing in the studied organelles was calculated using Manders' coefficient. Fraction of eGFP positive pixels overlapping with pixels positive for the organelle marker. Mean ± SD; *P* values are indicated; Kruskal–Wallis with Dunn's multiple comparisons test, PDI: *n* ≥ 6 cells, EEA1: *n* ≥ 14 cells, Rab11: *n* ≥ 12 cells. Representative experiment out of three independent experiments. [Colour figure can be viewed at wileyonlinelibrary.com]

(Mock) ($P < 0.001$) (Fig. 3B). Recently, it was shown that CaM can associate with rabbit TRPV5 in its non-calcified apo state (Zuidscherwoude et al., 2023). To determine whether the colocalisation of CaM with human TRPV5 depends on the calcification of CaM, HEK293 were transfected with eGFP-hTRPV5 and mCherry-tagged CaM1234 (Fig. 3A). This CaM mutant is deficient in binding $Ca^{2+}$ because of point mutations in all four EF hands. CaM1234 co-localised with hTRPV5 ($P < 0.001$) and the degree of co-localisation was not different to that of WT CaM and hTRPV5 (Fig. 3B).

## The effect of CaM on TRPV5 ER exit

To understand the function of the intracellular CaM–TRPV5 interaction, we investigated whether CaM influences TRPV5 trafficking through the secretory pathway. In HEK293 TET-off cells, the intracellular location of pulse-expressed TRPV5 was investigated over time in relation to ER marker calnexin. In accordance with results described in Fig. 2, TRPV5 co-localised with the ER 10 h after pTRE-hTRPV5 and Mock co-transfection, after which TRPV5 adopted a vesicular localisation pattern (Fig. 4A). The fraction of TRPV5 overlapping with the ER decreased significantly over time ($t = 0$ vs. $t = 4$, $P = 0.00759$; $t = 0$ vs. $t = 8$, $P < 0.001$) (Fig. 4D).

Cells co-transfected with WT CaM displayed a different intracellular trafficking of TRPV5, in which TRPV5 remained associated with the ER for a longer period (Fig. 4B). The fraction of TRPV5 co-occurring with calnexin at $t = 4$ did not significantly differ from the fraction at $t = 0$ (Fig. 4D). Importantly, the CaM-induced retention of TRPV5 within the ER was dependent on the calcification status of CaM. TRPV5 co-expressed with CaM1,2,3,4 exited the ER in a similar time frame as TRPV5 in the control cells (Fig. 4C and D). The inhibitory effect of CaM WT overexpression on TRPV5 forward trafficking was evident at timepoint $t = 4$. In CaM WT co-expressing cells, a larger fraction of TRPV5 was co-localising with calnexin compared to TRPV5 in mock-transfected control cells ($P = 0.0160$) and cells with CaM1,2,3,4 co-expression ($P < 0.001$) (Fig. 4D).

## TRPV5 expression of cells treated with CaM siRNA

To confirm the role of CaM in TRPV5 protein expression, HEK293 cells were transfected with three different siRNA pools, to target the three independent CaM genes (*CALM1–3*) or non-targeting (NT) control siRNA. Subsequently, these cells were transfected with eGFP-tagged hTRPV5. The knockdown efficiency of CaM and TRPV5 expression levels were determined by western blotting

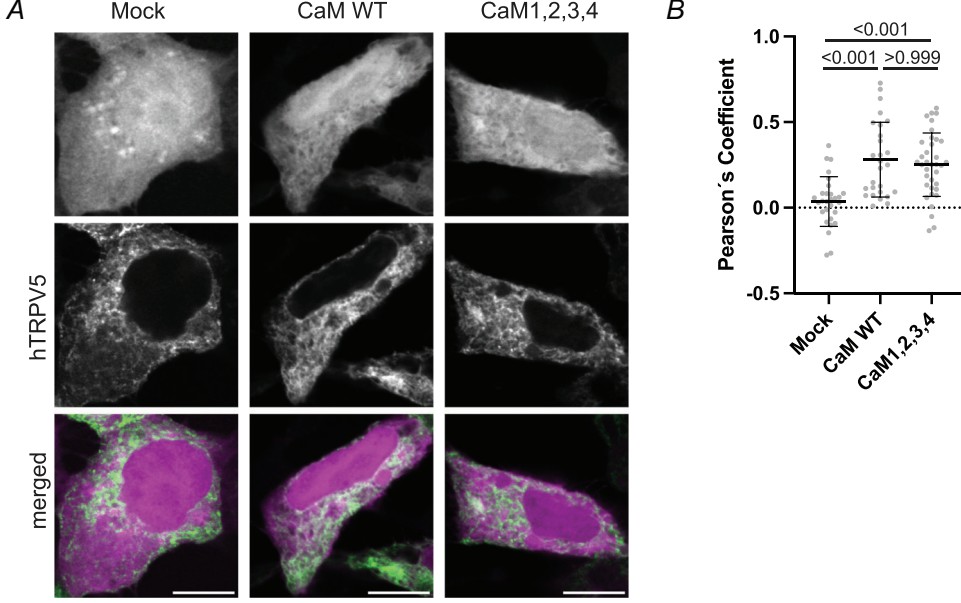

**Figure 3. $Ca^{2+}$-independent intracellular co-localisation of CaM with TRPV5**
*A*, representative confocal microscopy images of HEK293 cells transfected with eGFP-hTRPV5 and mCherry (mock), mCherry-CaM WT or mCherry-CaM1,2,3,4. Bottom row displays merged images of eGFP-hTRPV5 (green) with mCherry or mCherry-tagged CaM variant (magenta). Scale bar = 10 µm. *B*, the co-localisation of eGFP-hTRPV5 and mCherry, mCherry-CaM WT or mCherry-CaM1,2,3,4 was analysed using the Pearson's coefficient. Correlation of eGFP and mCherry intensities per pixel in cellular areas excluding the nucleus. A Pearson's coefficient of 0, indication of no co-localisation, is shown as the dotted line. Mean ± SD; *P* values for the Kruskal–Wallis with Dunn's multiple comparisons test are indicated; Wilcoxon signed rank test with a theoretical mean of 0.000; Mock: *P* = 0.151, CaM WT: *P* < 0.001, CaM 1,2,3,4: *P* < 0.001, *n* ≥ 27 cells; data from three independent experiments. [Colour figure can be viewed at wileyonlinelibrary.com]

of cell lysates (Fig. 5*A*, top). CaM siRNA treated cells retained a residual 30% of CaM protein expression, which is a significantly lower level compared to NT siRNA treated cells ($P < 0.001$) (Fig. 5*B*). This reduction in CaM protein expression was sufficient to elicit an effect on the protein levels of TRPV5. Compared to NT siRNA treated cells, CaM knockdown HEK293 cells contained significantly less TRPV5 protein ($P = 0.0456$) (Fig. 5*B*).

Within the same experiment, we also investigated whether the decreased TRPV5 expression in CaM knockdown cells resulted in a lower abundance of TRPV5 channels on the cell surface by a biotinylation assay. HEK293 cells treated with siRNA and transfected with eGFP-hTRPV5 were incubated with biotin to label and precipitate their cell surface proteins. The fraction of TRPV5 on the membrane was subsequently visualised by western blotting (Fig. 5*A*, bottom). There appears to be reduced complex glycosylation of membrane TRPV5 in cells with a knockdown of CaM, suggesting a disturbed secretory pathway. Interestingly, the amount of TRPV5 on

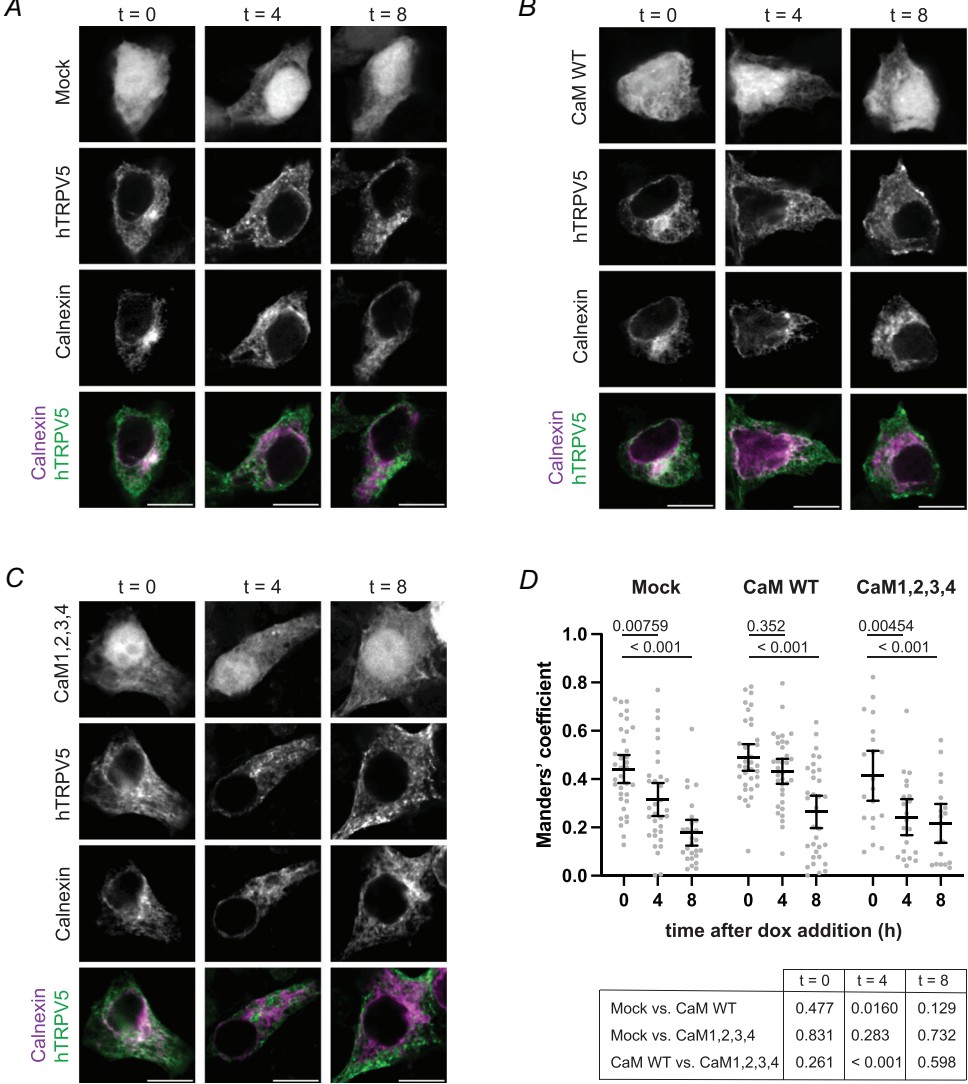

**Figure 4. TRPV5 retention in the ER upon CaM overexpression**
Representative confocal microscopy images of HEK293 Tet-off cells transfected with pTRE3G-eGFP-hTRPV5, genetic ER marker calnexin-mIFP and mCherry (mock, *A*), mCherry-CaM (WT, *B*) or mCherry-CaM1,2,3,4 (*C*). The transcription of pTRE3G-eGFP-hTRPV5 was stopped after 10 h by doxycycline addition. Cells were fixed immediately ($t = 0$), 4 or 8 h after doxycycline addition ($t = 4$ and $t = 8$, respectively). Bottom row displays merged images of eGFP-hTRPV5 in green and ER marker calnexin in magenta. Scale bar = 10 μm. *D*, fraction of eGFP-hTRPV5 residing in the ER was calculated using Manders' coefficient. Fraction of eGFP positive pixels overlapping with pixels positive for calnexin-mIFP. Mean with 95% confidence interval; two-way ANOVA with Tukey's multiple comparisons test; *P* values are indicated within the figure (within CaM variant) or in the table below (within time point); $N \geq 19$ cells, data from three independent experiments. [Colour figure can be viewed at wileyonlinelibrary.com]

the membrane of CaM siRNA-treated cells did not differ from NT siRNA-treated cells (Fig. 5*B*). Moreover, when correcting for the total TRPV5 expression levels, a higher fraction of TRPV5 was found on the plasma membrane of CaM knockdown cells ($P = 0.00453$) (Fig. 5*C*).

### Membrane expression of TRPV5 mutants affecting CaM binding

To determine the effect of CaM–TRPV5 interaction on TRPV5 trafficking, cells were transfected with HA-tagged rbTRPV5 WT, or with a TRPV5 mutant where the mutated site is involved in CaM binding. Although the T709A mutation is able to bind CaM to a similar extent as WT TRPV5, TRPV5 W702A, R706E and C-terminal truncation mutant 698X all show impaired CaM binding (de Groot et al., 2011; van der Wijst et al., 2017). The total

TRPV5 protein abundance is not significantly different between WT TRPV5 and TRPV5 mutants W702A, R706E and T709A and C-terminal truncation mutant 698X (Fig. 6*A*, upper, and 6*B*, upper). However, the TRPV5 variants differ from each other when considering their plasma membrane abundance ($P < 0.001$) (Fig. 6*A*, lower, and 6*B*, middle). Disrupted CaM–TRPV5 binding resulted in an increased TRPV5 expression on the plasma membrane. Compared to WT, TRPV5 698X has a higher abundance on the plasma membrane ($P = 0.0393$).

### Intracellular Ca$^{2+}$ concentration of CaM siRNA-treated TRPV5 positive cells

Next, Fura-2 Ca$^{2+}$ imaging experiments were performed to assess the effect of CaM on human TRPV5 function. This readout of the intracellular Ca$^{2+}$ concentration

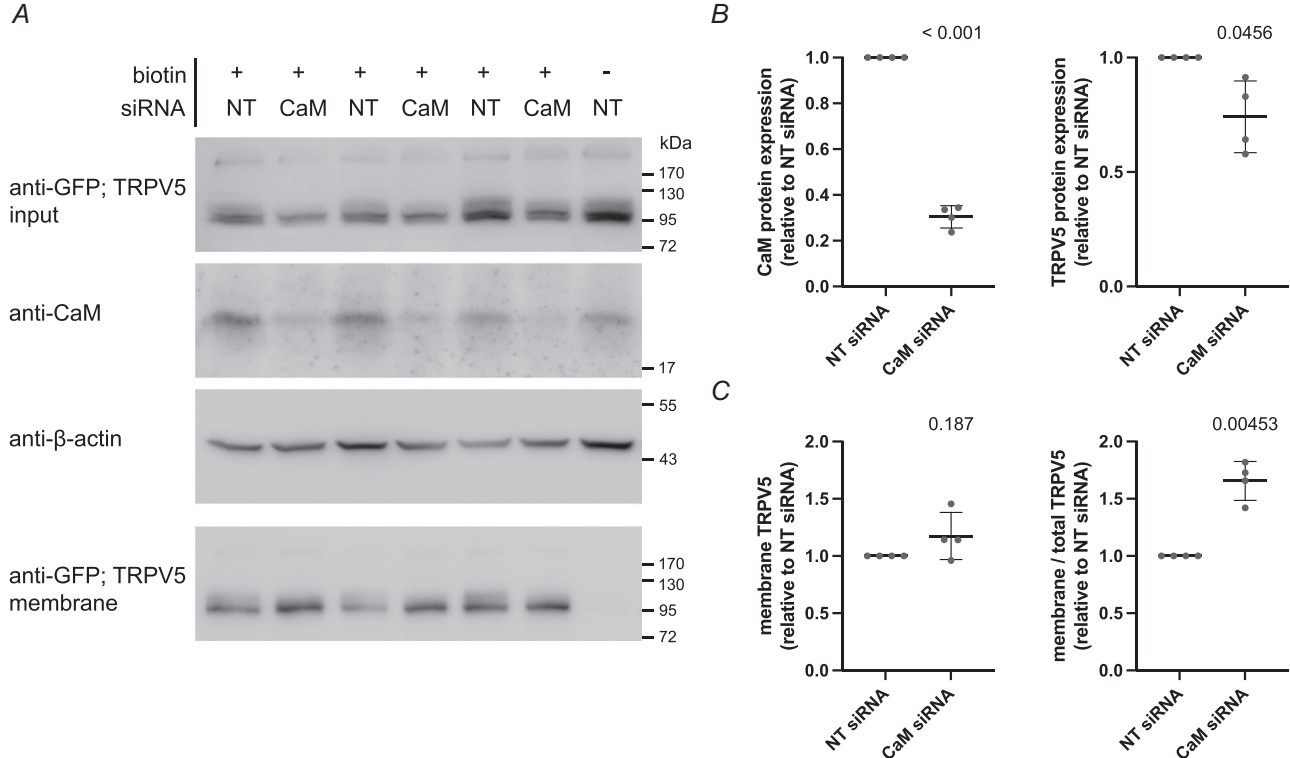

**Figure 5. Effect of CaM KD on expression and membrane abundance of TRPV5**
HEK293 cells treated with a pool of siRNAs targeting *CALM1*, *CALM2* and *CALM3* or NT control siRNA and subsequently transfected with eGFP-hTRPV5 24 h later. Cell surface proteins were biotinylated. *A*, western blot of total cell lysates, probed for eGFP-hTRPV5 using anti-GFP antibodies, endogenous CaM and *β*-actin, and a western blot of precipitated cell surface proteins probed for cell surface eGFP-hTRPV5 with anti-GFP antibodies (bottom). The control without biotin showed no precipitated eGFP-hTRPV5. Representative experiment (with triplicates) out of four independent experiments. *B*, quantification of CaM and TRPV5 protein expression. Protein band intensities corrected for loading by *β*-actin. Data points represent the average intensity of triplicates within each independent experiment ($n = 4$), normalised to the NT siRNA controls. Mean ± SD; one sample *t* test with a theoretical mean of 1; *P* values are indicated. *C*, quantification of cell surface hTRPV5 expression. Protein band intensities corrected for loading by *β*-actin (left) or corrected for total hTRPV5 protein (right). Data points represent the average intensity of triplicates within each independent experiment ($n = 4$), normalised to the NT siRNA controls. Mean ± SD; one sample *t* test with a theoretical mean of 1; *P* values are indicated.

represents a combination of the direct inhibition capabilities of CaM and its effect on the number of TRPV5 channels on the plasma membrane. CaM knockdown cells and cells treated with NT control siRNA were subsequently transfected with eGFP-hTRPV5 and either mCherry (CaM siRNA) or mIFP (NT siRNA) fluorescent protein. In the Fura-2 experimental setup, these cells were mixed to directly compare the $Ca^{2+}$ response in CaM siRNA- and NT siRNA-treated cells (Fig. 7A). TRPV5-transfected cells showed a peak response with plateau upon addition of $Ca^{2+}$ (Fig. 7B). CaM siRNA-treated cells had on average a higher Fura-2

ratio at the start of the experiment, as well as at the peak of the $Ca^{2+}$ response, compared to NT siRNA treated cells (start, $P < 0.001$; peak, $P = 0.0174$) (Fig. 7C), showing that a knockdown of CaM results in a higher intracellular $Ca^{2+}$ concentration.

## CaM regulation of trafficking of the $Ca^{2+}$ impermeable hTRPV5 D542A mutant

Lastly, we investigated the role of these disturbed intracellular calcium concentrations on TRPV5 trafficking. The

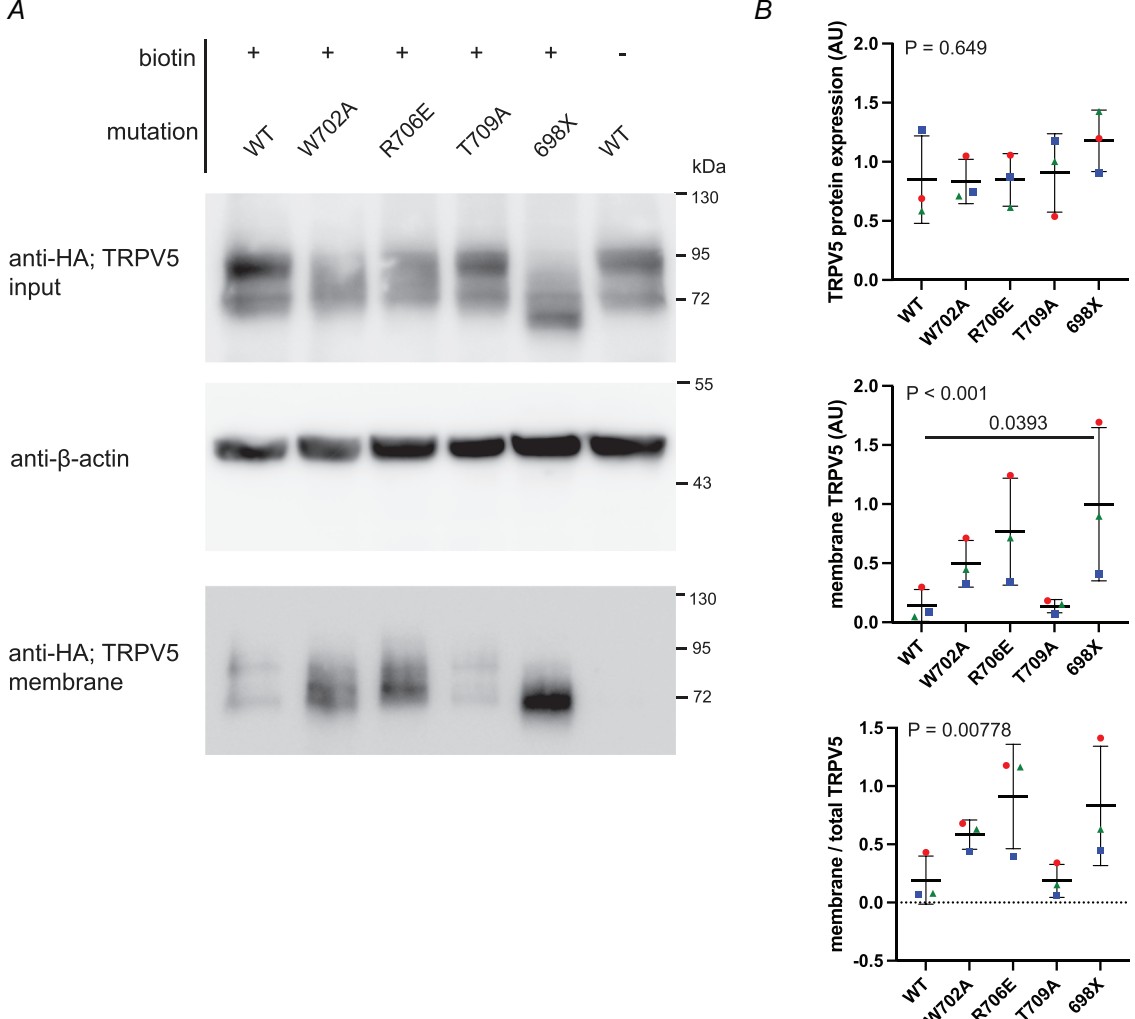

**Figure 6. The plasma membrane abundance of TRPV5 mutants impaired in CaM binding**
HEK293 cells were transfected with HA-tagged rabbit-TRPV5 and indicated mutants. Twenty-four hours later cell surface proteins were biotinylated. A, western blot of total cell lysates, probed for TRPV5 using anti-HA antibodies and for $\beta$-actin, and a western blot of precipitated cell surface proteins probed for cell surface TRPV5 with anti-HA antibodies (lower). The control without biotin showed no precipitated TRPV5. Representative experiment (with triplicates) out of three independent experiments. B, quantification of TRPV5 protein expression (upper). Protein band intensities corrected for loading by $\beta$-actin. Quantification of cell surface TRPV5 expression corrected for loading by $\beta$-actin (middle) or corrected for total TRPV5 protein (lower). Data points are colour coded based on experiment number. Mean $\pm$ SD; Friedman test with Dunn's multiple comparisons test with WT as control group; P values are indicated. [Colour figure can be viewed at wileyonlinelibrary.com]

canonical role of CaM in blocking superfluous $Ca^{2+}$ cell entry through TRPV5 channels is disrupted when the expression of CaM is downregulated. The resulting high intracellular $Ca^{2+}$ levels could affect TRPV5 trafficking via a $Ca^{2+}$-dependent trafficking pathway (independent of TRPV5-CaM interaction in the secretory pathway). To address this, we have generated a hTRPV5 expression construct containing a mutation at the TRPV5s selectivity filter, D542A (Nilius, Vennekens, et al., 2001). The $Ca^{2+}$ permeability of hTRPV5 D542A was assessed by the Fura-2 assay. eGFP-hTRPV5 WT and eGFP-hTRPV5 D542A transfected cells were co-transfected with a marker fluorescent protein and subsequently mixed to analyse their $Ca^{2+}$ response simultaneously. Although the intracellular concentrations at baseline are not different between the TRPV5 variants, hTRPV5 WT is able

to respond to addition of extracellular $Ca^{2+}$ whereas hTRPV5 D542A is not ($P < 0.001$) (Fig 8*A*).

We subjected eGFP-hTRPV5 D542A expressing cells to the biotinylation assay to assess the effect of CaM knockdown on TRPV5 trafficking, independent of changes in the intracellular $Ca^{2+}$ concentrations. HEK293 cells treated with siRNA and transfected with eGFP-hTRPV5 D542A were incubated with biotin to label and to precipitate their cell surface proteins. The fraction of hTRPV5 on the membrane was subsequently visualised by western blotting (Fig. 8*B*). hTRPV5 D542A channels are able to reach the cell surface. The total abundance of hTRPV5 D542A protein is elevated in CaM siRNA-treated cells compared to the NT siRNA-treated cells ($P = 0.0100$) (Fig. 8*B*, upper panel, and 8*C*, right) indicative of disrupted protein trafficking. When

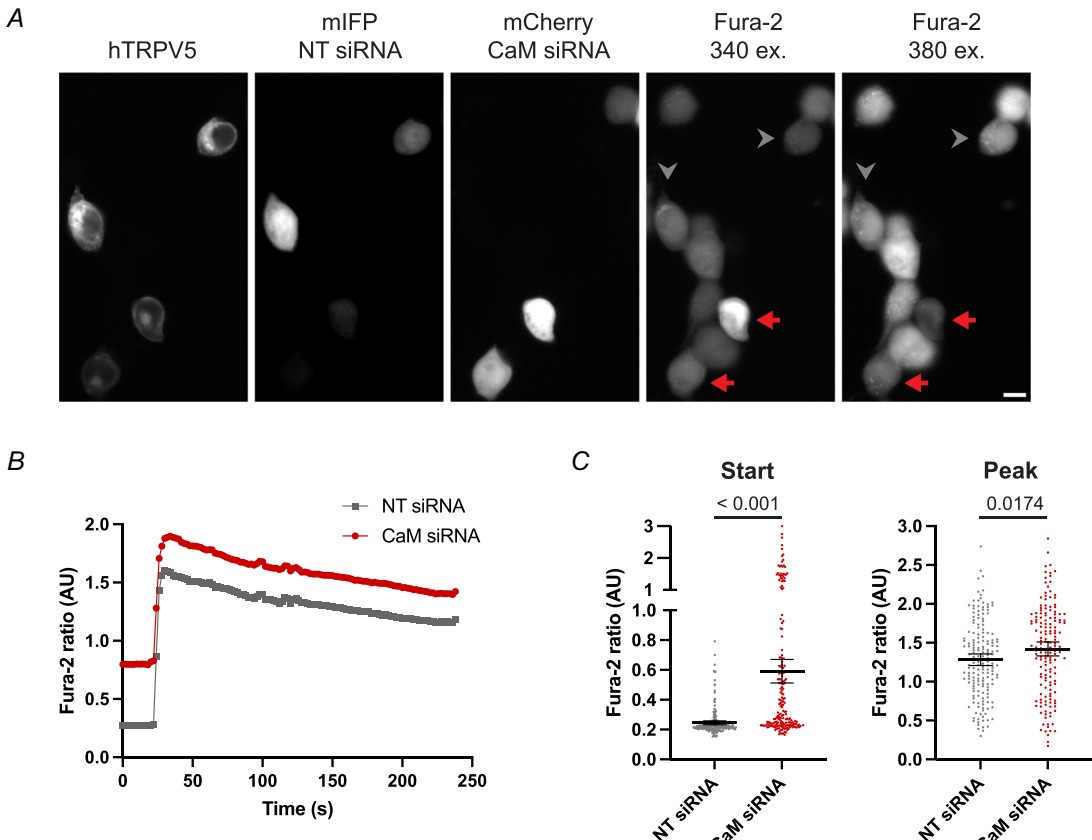

**Figure 7. Effect of CaM KD on the intracellular calcium concentration**
HEK293 cells treated with a CaM siRNA and transfected with eGFP-hTRPV5 and mCherry were mixed with HEK293 cells treated with NT control siRNA and transfected with eGFP-hTRPV5 and mIFP. Cells were loaded with Fura-2 and subsequently incubated with EDTA, after which the intracellular $Ca^{2+}$ concentration upon $Ca^{2+}$ addition was monitored in a live-cell imaging experiment. *A*, still images before $Ca^{2+}$ addition. NT siRNA treated cells were identifiable by mIFP positivity (grey arrowheads) and CaM siRNA treated cells by mCherry positivity (red arrows). Scale bar = 10 μm. *B*, representative experiment of intracellular $Ca^{2+}$ measurements over time with addition of buffer containing $Ca^{2+}$ at 20 s. Datapoints depict the average Fura-2 ratio (340/380 nm excitation) of 19 NT siRNA cells and 19 CaM siRNA cells that were simultaneously studied in one dish. *C*, Fura-2 ratio of individual cells before addition of $Ca^{2+}$ containing buffer (left) or max Fura-2 ratio (right). Mean with 95% confidence interval; *P* values are indicated; Mann–Whitney test; number of cells ≥ 163 from eight experiments. [Colour figure can be viewed at wileyonlinelibrary.com]

correcting for these total TRPV5 D542A expression levels, the fraction of TRPV5 D542A on the plasma membrane is not significantly different between CaM knockdown and control cells (Fig. 8*D*).

## Discussion

The present study has uncovered a novel mechanism by which CaM controls TRPV5. CaM slowed down TRPV5 trafficking through the secretory pathway and

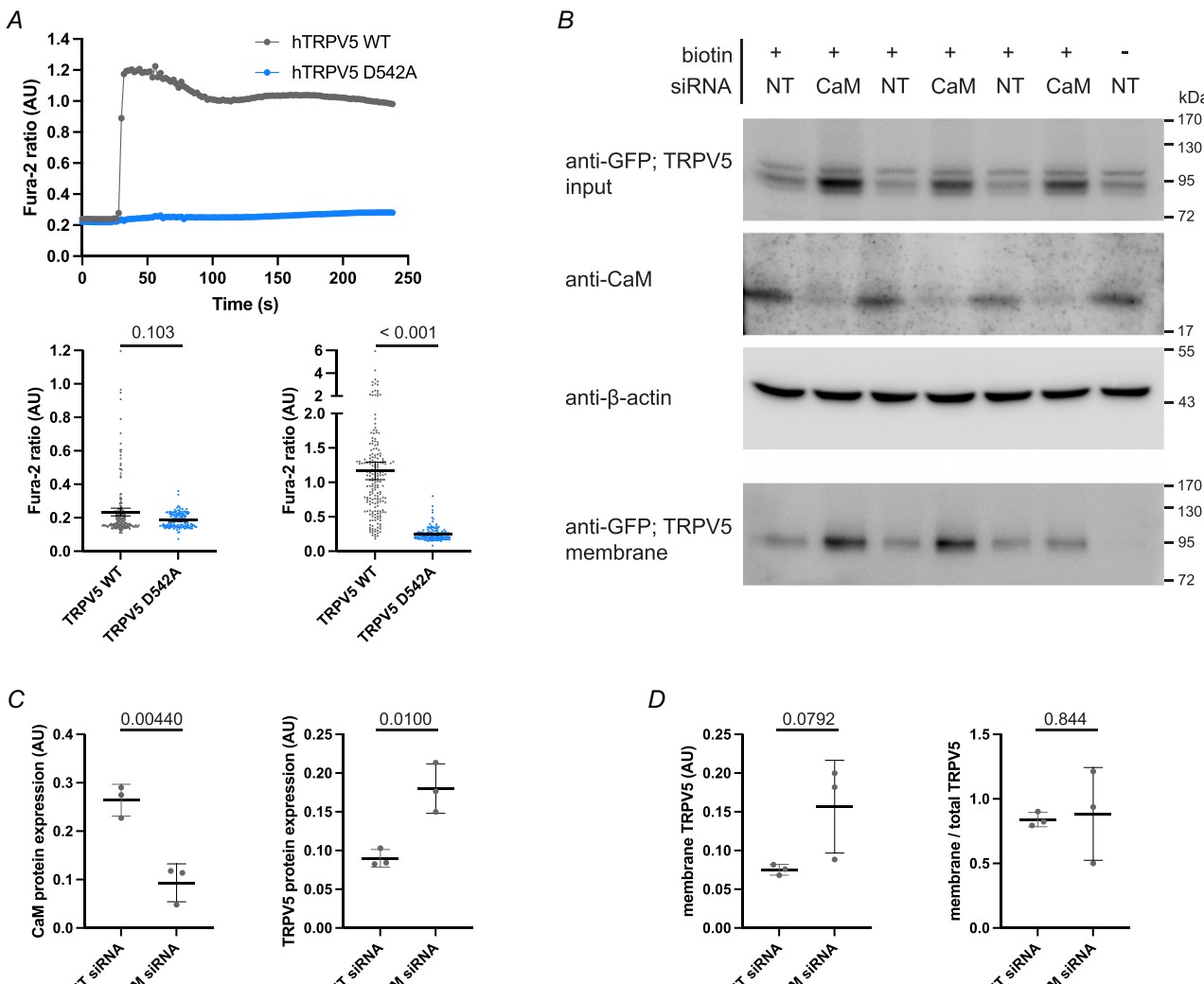

**Figure 8. Effect of CaM KD on expression and membrane abundance of hTRPV5 D542A**

*A*, HEK293 cells co-transfected with eGFP-hTRPV5 WT and mIFP were mixed with cells co-transfected with eGFP-hTRPV5 D542A and mCherry. Cells were loaded with Fura-2 and subsequently incubated with EDTA, after which the intracellular $Ca^{2+}$ concentration upon $Ca^{2+}$ addition was monitored in a live-cell imaging experiment. Upper: representative experiment of intracellular $Ca^{2+}$ measurements over time with addition of buffer containing $Ca^{2+}$ at 20 s. Datapoints depict the average Fura-2 ratio (340/380 nm excitation) of 37 TRPV5 WT cells and 59 TRPV5 D542A cells. Lower: Fura-2 ratio of individual cells before addition of $Ca^{2+}$ containing buffer (left) or max Fura-2 ratio (right). Mean with 95% confidence interval; *P* values are indicated; Mann–Whitney test; number of cells $\geq$ 149 from three experiments. *B*, HEK293 cells treated with a pool of siRNAs targeting *CALM1*, *CALM2* and *CALM3* or NT control siRNA and subsequently transfected with eGFP-hTRPV5 D542A 24 h later. Cell surface proteins were biotinylated. Western blot of total cell lysates, probed for eGFP-hTRPV5 D542A using anti-GFP antibodies, endogenous CaM and *β*-actin, and a western blot of precipitated cell surface proteins probed for cell surface eGFP-hTRPV5 D542A with anti-GFP antibodies (bottom). The control without biotin showed no precipitated eGFP-hTRPV5 D542A. Representative experiment of three experiments. *C*, quantification of CaM and hTRPV5 D542A protein expression. Protein band intensities corrected for loading by *β*-actin. Mean ± SD; *t* test; *P* values are indicated. *D*, quantification of cell surface hTRPV5 D542A expression. Protein band intensities corrected for loading by *β*-actin (left) or corrected for total hTRPV5 protein (right). Mean ± SD; *P* values are indicated. [Colour figure can be viewed at wileyonlinelibrary.com]

limited the fraction of TRPV5 channels that localises on the cell surface. Similar to its role in direct inhibition of TRPV5 at the single channel level, CaM regulated the forward trafficking of TRPV5 in a $Ca^{2+}$-dependent manner.

The activity of TRPV5 needs to be tightly controlled to maintain $Ca^{2+}$ homeostasis. In DCT and CNT cells, a major fraction of TRPV5 is stored in intracellular compartments below the apical membrane, suggesting that channel trafficking is a key mechanism in controlling the rate of active $Ca^{2+}$ transport (Loffing et al., 2001; van de Graaf et al., 2006; Wu et al., 2011). In accordance, when TRPV5 is overexpressed in cell lines, plasma membrane localised channels are not discernible via immuno-cytochemistry and can only be studied with sensitive techniques, such as cell surface biotinylation (Hellwig et al., 2005; Lambers et al., 2007; van de Graaf et al., 2006, 2008). Exogenously expressed TRPV5 is predominantly found in intracellular vesicular compartments, whereas here we show that, when overexpressed together with CaM in HEK293 cells, a large fraction of TRPV5 is localised within the ER. To prevent overexpression artefacts, we employed the HEK293 Tet-off cell line to limit the transcription time of TRPV5 in the context of endogenous CaM expression. Over time, the subcellular location of TRPV5 shifted from the ER to intracellular vesicles. This is in line with the hypothesis that TRPV5 channels are continuously endocytosed and stored below the apical membrane of DCT and CNT cells, allowing for a quick response upon cues for an increased $Ca^{2+}$ reabsorption, by translocating open channels into the plasma membrane (van de Graaf et al., 2006, 2008).

The fate of newly synthesised TRPV5 protein was previously shown in a study detecting [$^{35}$S]methionine/cysteine labelled rbTRPV5 in a pulse-chase analysis. Interestingly, a fraction of rbTRPV5 acquired complex glycosylation within 1 h after protein translation, and, thereafter, the ratio between complexly glycosylated and core glycosylated TRPV5 remained similar for more than 8 h, suggesting that TRPV5 is intracellularly stored in different glycosylation states (van de Graaf et al., 2008). This raises the question of whether, next to internalisation and recycling, there could be another mechanism in play to regulate the plasma membrane abundance of TRPV5. Here, we show that newly synthesised hTRPV5 channels are retained in the ER before they reach the plasma membrane.

The ER is involved in many essential cellular processes, including protein and lipid synthesis and $Ca^{2+}$ storage. It consists of interconnected membrane sheets and tubules, the latter forming a reticular network in perinuclear and peripheral regions of the cell. ER morphology is linked to function and can vary in different cell types (Schwarz & Blower, 2016). By contrast to, for example, proximal tubule cells, DCT cells contain scarcely any ER sheets but do possess a network of ER tubules throughout the cytoplasm (Bergeron et al., 1987).

With a more than 1000-fold higher $Ca^{2+}$ concentration in the ER lumen compared to the cytoplasm, the assembly of TRPV5 channels in the ER comes with the danger of $Ca^{2+}$ leakage (Taylor et al., 2009). The requirement for $PI(4,5)P_2$ probably silences TRPV5 until it reaches the plasma membrane. In the secretory and endocytic pathway, a low pH comes into play, inhibiting TRPV5 by altering $PI(4,5)P_2$ binding to the channel (Fluck et al., 2022). Because the pH in the ER lumen is neutral, it is interesting to speculate whether (apo)CaM, prebound to TRPV5 in the ER, could serve as an additional safeguard to prevent $Ca^{2+}$ leaks from the ER. The interaction between TRPV5 and CaM has been extensively studied (Dang et al., 2019; Hughes, Pumroy et al., 2018; Kovalevskaya et al., 2012); however, information on the behaviour of the full-length proteins in intact subcellular compartments is sparse. Recently, it was shown by fluorescence life-time imaging that both CaM and apoCaM associated with TRPV5 in the plasma membrane. In addition, CaM and the $Ca^{2+}$-insensitive mutant CaM1,2,3,4 were found to interact with TRPV5 in the intracellular compartment (Zuidscherwoude et al., 2023). Here, we show that TRPV5 was co-localising with CaM WT and CaM1,2,3,4 intracellularly. Because hTPRV5 was mainly present in the ER and CaM WT regulated its ER exit speed, we can assume that CaM and TRPV5 interacted in the ER.

CaM has been implicated in the forward trafficking and cell surface expression of other ion channels (Biswas et al., 2008; Etxeberria et al., 2008; Lee et al., 2003; Rosenbaum et al., 2004; Wang et al., 2007). However, there is no consensus on the $Ca^{2+}$ dependency of this process, nor a general binding site for CaM to exert this specific action. This indicates that CaM is regulating the plasma membrane expression of ion channels in a tailored manner. Here, we have uncovered a novel mechanism by which the abundance of TRPV5 channels on the cell surface can be controlled in accordance with the intra-cellular $Ca^{2+}$ concentrations by the actions of $Ca^{2+}$ sensor CaM. Interestingly, factors from the extracellular space could interfere with the intracellular trafficking of TRPV5 by regulating CaM binding. For example, parathyroid hormone signalling and PAR-1 signalling by plasmin increases the phosphorylation of a CaM binding site in TRPV5, thereby altering CaM–TRPV5 interaction (de Groot et al., 2011; Tudpor et al., 2012).

The lifetime of a TRPV5 channel is complex, involving the secretory pathway, plasma membrane localisation, recycling and degradation. Studying regulators of TRPV5 activity is challenging as aberrant TRPV5 mediated $Ca^{2+}$ cell entry can lead to cytotoxicity potentially skewing results. Moreover, certain assays, such as the Fura-2 assay, cannot discern whether heightened intracellular $Ca^{2+}$ concentrations have arisen from an augmented surface

abundance of TRPV5 or from a disrupted inactivation of TRPV5. Therefore, for future studies into TRPV5 regulation, we suggest that multiple assays are employed to investigate both TRPV5 membrane abundance and channel activity to accurately determine mode of actions.

In conclusion, the present study demonstrates a new role for CaM in regulating TRPV5 mediated $Ca^{2+}$ entry. Next to its ability to directly block the TRPV5 channel pore in a $Ca^{2+}$-dependent negative feedback mechanism; here, we show that CaM controlled TRPV5 trafficking in the secretory pathway, thereby regulating the membrane abundance of this ion channel.

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

## Additional information

### Data availability statement

The data available in the article and the source data are available from the corresponding author upon reasonable request.

### Competing interests

The authors declare that they have no competing interests.

### Author contributions

M.Z. and J.v.W. conceived and designed the experiments. M.Z., B.v.d.L. and G.W. performed the experiments. M.Z., B.v.d.L., G.W. and J.G.H. analysed and interpreted the data. M.Z., T.G. and J.G.H. wrote the manuscript, with critical review and approval by all authors. J.v.W. and J.G.H. supervised the work. All authors have approved the final version of the manuscript submitted for publication and agree to be accountable for all aspects of the work. All persons designated as authors qualify for authorship and all those who qualify for authorship are listed.

### Funding

This study was financially supported by the Dutch Organization for Scientific Research (NWO) (ENW OC ENW KLEIN 2022/ENW/01277767).

### Keywords

calcium homeostasis, calmodulin, channel regulation, secretory pathway, TRP channel

## Supporting information

Additional supporting information can be found online in the Supporting Information section at the end of the HTML view of the article. Supporting information files available:

**Peer Review History**

