## [Peer Review History · The Journal of Physiology]

Calmodulin regulates TRPV5 intracellular trafficking and plasma membrane abundance

Malou Zuidscherwoude, Teodora Grigore, Brenda van de Langenberg, Guusje Witte, Jenny van der Wijst, and Joost G Hoenderop

DOI: 10.1113/JP286182

Corresponding author(s): Joost Hoenderop (joost.hoenderop@radboudumc.nl)

Review Timeline:

Submission Date:	22-Dec-2023
Editorial Decision:	09-Feb-2024
Revision Received:	15-Oct-2024
Accepted:	29-Oct-2024

Senior Editor: Peking Fong

Reviewing Editor: Pawel Ferdek

Transaction Report:

Dear Dr Hoenderop,

Re: JP-RP-2023-286182 "Calmodulin regulates TRPV5 intracellular trafficking and plasma membrane abundance" by Malou Zuidscherwoude, Teodora Grigore, Brenda van de Langenberg, Guusje Witte, Jenny van der Wijst, and Joost G Hoenderop

Thank you for submitting your manuscript to The Journal of Physiology. It has been assessed by a Reviewing Editor and by 2 expert referees and we are pleased to tell you that it is potentially acceptable for publication following satisfactory major revision.

REVISION CHECKLIST:

We look forward to receiving your revised submission.

Yours sincerely,

Peying Fong
Senior Editor
The Journal of Physiology

REQUIRED ITEMS

- Author photo and profile. First or joint first authors are asked to provide a short biography (no more than 100 words for one author or 150 words in total for joint first authors) and a portrait photograph. These should be uploaded and clearly labelled together in a Word document with the revised version of the manuscript. See Information for Authors for further details.

- Your manuscript must include a complete Additional Information section, including competing interests; funding; author contributions and acknowledgements.

- Please upload separate high-quality figure files via the submission form.

- You must upload original, uncropped western blot/gel images (including controls) if they are not included in the manuscript. This is to confirm that no inappropriate, unethical or misleading image manipulation has occurred. These should be uploaded as 'Supporting information for review process only'. Please label/highlight the original gels so that we can clearly see which sections/lanes have been used in the manuscript figures. For more information, see: <https://physoc.onlinelibrary.wiley.com/hub/journal-policies#imagmanip>.

- Papers must comply with the Statistics Policy: https://jp.msubmit.net/cgi-bin/main.plex?form_type=display_requirements#statistics.

In summary:

- If $n \leq 30$, all data points must be plotted in the figure in a way that reveals their range and distribution. A bar graph with data points overlaid, a box and whisker plot or a violin plot (preferably with data points included) are acceptable formats.

- If $n > 30$, then the entire raw dataset must be made available either as supporting information, or hosted on a not-for-profit repository, e.g. FigShare, with access details provided in the manuscript.

- 'n' clearly defined (e.g. x cells from y slices in z animals) in the Methods. Authors should be mindful of pseudoreplication.

- All relevant 'n' values must be clearly stated in the main text, figures and tables.

- The most appropriate summary statistic (e.g. mean or median and standard deviation) must be used. Standard Error of the Mean (SEM) alone is not permitted.

- Exact p values must be stated. Authors must not use 'greater than' or 'less than'. Exact p values must be stated to three significant figures even when 'no statistical significance' is claimed.

- Please include an Abstract Figure file, as well as the Figure Legend text within the main article file. The Abstract Figure is a piece of artwork designed to give readers an immediate understanding of the research and should summarise the main conclusions. If possible, the image should be easily 'readable' from left to right or top to bottom. It should show the physiological relevance of the manuscript so readers can assess the importance and content of its findings. Abstract Figures should not merely recapitulate other figures in the manuscript. Please try to keep the diagram as simple as possible and without superfluous information that may distract from the main conclusion(s). Abstract Figures must be provided by authors no later than the revised manuscript stage and should be uploaded as a separate file during online submission labelled as File Type 'Abstract Figure'. Please also ensure that you include the figure legend in the main article file. All Abstract Figures should be created using BioRender. Authors should use The Journal's premium BioRender account to export high-resolution images. Details on how to use and access the premium account are included as part of this email.

EDITOR COMMENTS

Reviewing Editor:

Thank you for submitting your work to the Journal of Physiology. Your manuscript has been evaluated by two independent Referees. Your study proposing a novel role for calmodulin in the intracellular trafficking of TRPV5, has been identified as potentially interesting. However, there are several issues highlighted by the Referees that require your attention and response.

Furthermore, please pay attention to our Statistics policy (https://jp.msubmit.net/cgi-bin/main.plex?form_type=display_requirements#statistics). Of particular importance is the reporting of significant figures in p-values. Our guidelines are that for p-values greater than 0.001, the value should be reported to three significant figures (e.g. 0.00236, 0.523). This means there should be three digits reported, not including any leading zeros. For p-values less than 0.001, these can be reported as $P < 0.001$. We have noticed that some of your figures present values to three decimal places, rather than to three significant figures, which is not in alignment with our policy. We encourage the explicit statement of p-values in the main text and figures.

Senior Editor:

Thank you for submitting your manuscript for consideration by The Journal of Physiology. Overall, both Expert Referees and the Reviewing Editor note the high quality of the writing and remark on the interesting topic of TRPV5 trafficking and its regulation by calmodulin. Nonetheless, you will see that while all are convinced about the study's potential impactfulness, several concerns requiring attention were raised in the accompanying, detailed critiques.

As highlighted by Referee 1, because calmodulin has a documented previous impact on calcium uptake, it is important to establish that this does not indirectly exert effects on calmodulin's other proposed role in TRPV5 trafficking. Specificity of calmodulin's effects also are questioned. Both Referees suggest several additional experiments that might assist in clarifying the questions raised.

I recommend that you consider addressing Referee concerns completely and thank you in advance for doing so.

In addition, please see the Reviewing Editor's comments relating to our current Statistic Policy, requiring reporting of exact p values to 3 significant figures, rather than simply 3 places (there are several instances of the later within legends).

REFEREE COMMENTS

Referee #1:

In this study, Hoenderop and colleagues present new data showing that calmodulin (CaM) regulates TRPV5 intracellular trafficking and plasma membrane abundance. Overall, it is concluded that CaM not only directly inhibits TRPV5 channel-mediated calcium uptake, but also slows TRPV5 forward trafficking from the ER to the secretory pathway, thereby limiting the fraction of TRPV5 channels on the cell surface. And since a large fraction of TRPV5 is stored in intracellular compartments in the renal DCT and CNT, this trafficking is considered a key mechanism in controlling the rate of Ca^{2+} reuptake, allowing rapid recruitment of TRPV5 to the plasma membrane upon cues for increased Ca^{2+} reabsorption.

Major comments to be addressed:

1. To complete this project, it would be important to repeat the experiments using a TRPV5 phosphorylation mutant at threonine-709 of TRPV5. Given that the absence of CaM results in increased intracellular calcium levels, this could affect trafficking via a calcium-dependent trafficking pathway that is independent of CaM binding to TRPV5.

2. In parallel, removal of extracellular calcium should be tested using the immunobiological assay as this would be expected to have a similar effect. In the case of AQP0 (expressed in lens fiber cells), this protein is also regulated by trafficking to the plasma membrane in a similar way (<https://doi.org/10.1016/j.bbamem.2021.183853>). The authors should therefore consider and discuss this study as well. Removal of extracellular calcium increased the water permeability of AQP0 approximately fourfold. This Ca²⁺ sensitivity was shown to be dependent on calmodulin (CaM) with CaM inhibitors, thereby restoring water permeability. CaM also binds the C-terminus of AQP0 in a calcium-dependent manner and inhibits water permeability. Phosphorylation of an AQP0 C-terminal residue, Ser235 (a consensus site for PKA) also abolished CaM binding, and it was shown that phosphorylation of AQP0 is required for proper trafficking to the plasma membrane after biosynthesis. The authors should take this study into account and, as mentioned above, examine the role of threonine-709 to see if phosphorylation has a similar effect in TRPV5.

3. The effect of CaM knockdown on TRPV5 plasma membrane expression seems rather limited (Fig. 5), given that this is one of the main conclusions of this study. It seems that siRNA knockdown of CaM (targeting all 3 different CaM genes) leading to only 30% reduction in expression is insufficient to properly evaluate the effect of the lack of CaM on TRPV5 cell surface expression. Therefore a CRIPR knockout strategy would be more appropriate.

Minor comments:

4. The introduction is missing a paragraph stating that one of the main mechanisms of regulation of whole body calcium homeostasis via TRPV5 are via PTH, which increases the activity of TRPV5 via PKA-mediated phosphorylation at threonine-709 of TRPV5, thereby increasing the open probability of the channel by preventing CaM binding and channel inactivation. Although the current manuscript focuses on how CaM regulates TRPV5 trafficking and plasma membrane abundance, the above statement is important to provide a complete picture of the CaM regulatory pathways.

5. Another interesting study that the authors might mention in the paper is the one showing that Ca²⁺/CaM regulates cAMP-dependent CaV1.2 trafficking and plasma membrane targeting in hippocampal neurons, thereby contributing to activity-dependent gene expression (<https://doi.org/10.1523/JNEUROSCI.1720-07.2007>).

Referee #2:

The manuscript titled "Calmodulin regulates TRPV5 intracellular trafficking and plasma membrane abundance" submitted by Zuidschewoude et al. presents a compelling study uncovering a novel mechanism by which Calmodulin (CaM) controls the trafficking of TRPV5, a crucial ion channel involved in calcium homeostasis. The research demonstrates that CaM slows down TRPV5 trafficking through the secretory pathway, limiting its presence on the cell surface. This regulatory role is crucial for maintaining the tight control of TRPV5 activity necessary for calcium homeostasis. The study also delves into the intricate interplay between TRPV5 and CaM in the endoplasmic reticulum (ER), shedding light on the potential role of CaM as a safeguard against calcium leakage. The findings provide valuable insights into the tailored regulation of ion channel abundance on the cell surface in response to intracellular calcium concentrations. While the manuscript provides substantial contributions to the comprehension of CaM-mediated control of TRPV5, making it a commendable addition to the scientific literature in the field, it is recommended that the following questions and concerns be addressed:

1. Can the authors show if CaM-binding mutants of TRPV5, such as TRPV5-Δ696-729, TRPV5-W702A or TRPV5-R706E mutants have any difference in the ER retention compared to WT TRPV5?

2. Did the authors try a Calmodulin Inhibitory peptide to test if that prevents the retention of TRPV5 in the ER?

3. In the introduction section, it may be good to briefly mention that CaM is known to regulate the trafficking of several ion channels such as CaV1.2, KCNQ2, SK4/IK1 and others.

4. It is not the best idea to not talk anything about the CaM in section 1 and 2, given its co-expression in the presented data. Consider either reminding the rationale for using CaM in these experiments or presenting the data with cells that don't overly express CaM. Ideal would be to kick off with the localization of TRPV5 without the over-expression of CaM.

5. In Figure 6, it would be valuable for the authors to elaborate on how they discern whether the heightened intracellular Ca²⁺ in siCaM cells arises from an augmented surface abundance of CaM or from a potential failure in CaM-mediated inactivation of TRPV5. Addressing this aspect would enhance the depth of the discussion.

END OF COMMENTS

Confidential Review

22-Dec-2023

Rebuttal to Peer-Reviewer's comments

Manuscript:

JP-RP-2023-286182

Title:

Calmodulin regulates TRPV5 intracellular trafficking and plasma membrane abundance

Authors:

Malou Zuidscherwoude, Teodora Grigore, Brenda van de Langenberg, Guusje Witte, Jenny van der Wijst, Joost G. Hoenderop

Comment by the editors:

Reviewing Editor:

Thank you for submitting your work to the Journal of Physiology. Your manuscript has been evaluated by two independent Referees. Your study proposing a novel role for calmodulin in the intracellular trafficking of TRPV5, has been identified as potentially interesting. However, there are several issues highlighted by the Referees that require your attention and response. Furthermore, please pay attention to our Statistics policy (https://jp.msubmit.net/cgi-bin/main.plex?form_type=display_requirements#statistics). Of particular importance is the reporting of significant figures in p-values. Our guidelines are that for p-values greater than 0.001, the value should be reported to three significant figures (e.g. 0.00236, 0.523). This means there should be three digits reported, not including any leading zeros. For p-values less than 0.001, these can be reported as $P < 0.001$. We have noticed that some of your figures present values to three decimal places, rather than to three significant figures, which is not in alignment with our policy. We encourage the explicit statement of p-values in the main text and figures.

Senior Editor:

Thank you for submitting your manuscript for consideration by The Journal of Physiology. Overall, both Expert Referees and the Reviewing Editor note the high quality of the writing and remark on the interesting topic of TRPV5 trafficking and its regulation by calmodulin. Nonetheless, you will see that while all are convinced about the study's potential impactfulness, several concerns requiring attention were raised in the accompanying, detailed critiques.

As highlighted by Referee 1, because calmodulin has a documented previous impact on calcium uptake, it is important to establish that this does not indirectly exert effects on calmodulin's other proposed role in TRPV5 trafficking. Specificity of calmodulin's effects also are questioned. Both Referees suggest several additional experiments that might assist in clarifying the questions raised.

I recommend that you consider addressing Referee concerns completely and thank you in advance for doing so.

In addition, please see the Reviewing Editor's comments relating to our current Statistic Policy, requiring reporting of exact p values to 3 significant figures, rather than simply 3 places (there are several instances of the later within legends).

Thank you for considering our manuscript, we are pleased with the positive comments and the conclusion that the topic is of interest. We would like to thank the referees and editors for the excellent suggestions to make this study more impactful. We have added two new data figures to the manuscript. These address the potential indirect effects of calmodulin on TRPV5 trafficking via disturbed intracellular calcium concentrations and are substantiating the dependence of calmodulin – TRPV5 binding in the regulation of TRPV5 membrane abundance.

We have now also reported exact p-values in the text and figures in accordance with the Statistic Policy.

Comment by reviewer 1:

In this study, Hoenderop and colleagues present new data showing that calmodulin (CaM) regulates TRPV5 intracellular trafficking and plasma membrane abundance. Overall, it is concluded that CaM not only directly inhibits TRPV5 channel-mediated calcium uptake, but also slows TRPV5 forward trafficking from the ER to the secretory pathway, thereby limiting the fraction of TRPV5 channels on the cell surface. And since a large fraction of TRPV5 is stored in intracellular compartments in the renal DCT and CNT, this trafficking is considered a key mechanism in controlling the rate of Ca²⁺ reuptake, allowing rapid recruitment of TRPV5 to the plasma membrane upon cues for increased Ca²⁺ reabsorption.

We would like to thank the reviewer for her/his suggestions, as these have improved our manuscript. The specific comments are addressed below.

Major comments to be addressed:

1. To complete this project, it would be important to repeat the experiments using a TRPV5 phosphorylation mutant at threonine-709 of TRPV5. Given that the absence of CaM results in increased intracellular calcium levels, this could affect trafficking via a calcium-dependent trafficking pathway that is independent of CaM binding to TRPV5.

We have repeated biotinylation experiments using a panel of TRPV5 mutants that are previously shown to be sites of CaM binding. While the T709A mutation is able to bind CaM to a similar extent as WT TRPV5, TRPV5 W702A, R706E and C-terminal truncation mutant 698X all show impaired CaM binding (ref: doi: 10.1128/MCB.01319-10, doi: 10.1038/srep45489). We aimed to identify whether the regulatory effects of CaM on TRPV5 plasma membrane abundance is dependent on TRPV5-CaM interaction. The total TRPV5 protein abundance is not significantly different between WT TRPV5 and TRPV5 mutants W702A, R706E and T709A and C-terminal truncation mutant 698X (Figure 6A in revised manuscript). Compared to WT, TRPV5 mutants W702A, R706E and 698X have a higher abundance on the plasma membrane. (Due to the variation in effect size between the 3 independent experiments, only the abundance between WT and 698X is statistically significantly different in the post hoc analysis (Figure 6B in revised manuscript)). In line with the CaM siRNA experiments, this suggests that the lack of TRPV5-CaM interaction augments the abundance of TRPV5 on the plasma membrane.

To address a potential calcium-dependent effect on the TRPV5 trafficking pathway, independent of a direct CaM-TRPV5 interaction, we generated a human TRPV5 D542A mutant. This mutation in the selectivity filter of TRPV5 causes a disruption of the calcium flux through the TRPV5 channel (Figure 8A in revised manuscript). In these biotinylation experiments, the effect of CaM on TRPV5 protein abundance can be studied independent of disturbed intracellular calcium levels. We show in Figure 8B that hTRPV5 D542A protein abundance in the whole cell is higher when CaM expression is downregulated via siRNA. This indicates that CaM regulates the trafficking, recycling or stability of TRPV5 independent of changes in intracellular calcium levels.

2. In parallel, removal of extracellular calcium should be tested using the immunobiological assay as this would be expected to have a similar effect. In the case of AQP0 (expressed in lens fiber cells), this protein is also regulated by trafficking to the plasma membrane in a similar way (<https://doi.org/10.1016/j.bbamem.2021.183853>). The authors should therefore consider and discuss this study as well. Removal of extracellular calcium increased the water permeability of AQP0 approximately fourfold. This Ca²⁺ sensitivity was shown to be dependent on calmodulin (CaM) with CaM inhibitors, thereby restoring water permeability. CaM also binds the C-terminus of AQP0 in a calcium-dependent manner and inhibits water permeability. Phosphorylation of an AQP0 C-terminal residue, Ser235 (a consensus site for PKA) also abolished CaM binding, and it was shown that phosphorylation of AQP0 is required for proper trafficking to the plasma membrane after biosynthesis. The authors should take this study into account and, as mentioned above, examine the role of threonine-709 to see if phosphorylation has a similar effect in TRPV5.

We would like to thank the reviewer for bringing this review of relevant aquaporin studies to our attention. This review has been now included in the introduction section of the manuscript. Removal of extracellular calcium is not compatible with our assays, as the cells will detach from the plate surface. Therefore, we opted to study the behavior of the TRPV5 D542A mutant as discussed above.

3. The effect of CaM knockdown on TRPV5 plasma membrane expression seems rather limited (Fig. 5), given that this is one of the main conclusions of this study. It seems that siRNA knockdown of CaM (targeting all 3 different CaM genes) leading to only 30% reduction in expression is insufficient to properly evaluate the effect of the lack of CaM on TRPV5 cell surface expression. Therefore a CRIPR knockout strategy would be more appropriate.

The 60-70% reduction of CaM expression achieved in our study is commonly found in studies using CaM siRNA in mammalian cells, and it is hypothesized that the 3 different siRNA pools are competing for the RISC/Argonaute system (ref:<https://doi.org/10.1016/j.abb.2020.108680>).

CaM is an essential protein for cell survival. A CRISPR/Cas9-mediated gene deletion of CaM would be a cumbersome solution, as 3 different CaM genes will need to be targeted, while a

conditional CaM expression is introduced (refs: <https://doi.org/10.1016/j.ceca.2020.102207>, <https://doi.org/10.1074/jbc.M112.339382>).

We therefore opted for the CaM inhibitor W-7. With W-7 we aimed to target CaM expression completely for a limited amount of time.

Figure. Effect of W7 on TRPV5 expression.

The assay was performed 9 times (triplo's in 3 independent experiments) and the fraction of TRPV5 on the membrane (corrected for total TRPV5 expression) is plotted on the right.

Due to the inconsistency between the assay repeats, we are not confident that the W-7 inhibitor works for this specific assay in our hands. We have therefore decided not to include these results in the revised manuscript.

Minor comments:

4. The introduction is missing a paragraph stating that one of the main mechanisms of regulation of whole body calcium homeostasis via TRPV5 are via PTH, which increases the activity of TRPV5 via PKA-mediated phosphorylation at threonine-709 of TRPV5, thereby increasing the open probability of the channel by preventing CaM binding and channel inactivation. Although the current manuscript focuses on how CaM regulates TRPV5 trafficking and plasma membrane abundance, the above statement is important to provide a complete picture of the CaM regulatory pathways.

We thank the reviewer for this suggestion to make the introduction more comprehensive. We have added a section dedicated to the role of PTH on calcium homeostasis via CaM in the introduction.

5. Another interesting study that the authors might mention in the paper is the one showing that Ca²⁺/CaM regulates cAMP-dependent CaV1.2 trafficking and plasma membrane targeting in hippocampal neurons, thereby contributing to activity-dependent gene expression (<https://doi.org/10.1523/JNEUROSCI.1720-07.2007>).

We would like to thank the reviewer for bringing this relevant CaV1.2 study to our attention. This study has been now included in the introduction section of the manuscript.

Comment by reviewer 2:

The manuscript titled "Calmodulin regulates TRPV5 intracellular trafficking and plasma membrane abundance" submitted by Zuidsherwoude et al. presents a compelling study uncovering a novel mechanism by which Calmodulin (CaM) controls the trafficking of TRPV5, a crucial ion channel involved in calcium homeostasis. The research demonstrates that CaM slows down TRPV5 trafficking through the secretory pathway, limiting its presence on the cell surface. This regulatory role is crucial for maintaining the tight control of TRPV5 activity necessary for calcium homeostasis. The study also delves into the intricate interplay between TRPV5 and CaM in the endoplasmic reticulum (ER), shedding light on the potential role of CaM as a safeguard against calcium leakage. The findings provide valuable insights into the tailored regulation of ion channel abundance on the cell surface in response to intracellular calcium concentrations. While the manuscript provides substantial contributions to the comprehension of CaM-mediated control of TRPV5, making it a commendable addition to the scientific literature in the field, it is recommended that the following questions and concerns be addressed:

We would like to thank the reviewer for the positive comments and suggestions for experiments to make the study more comprehensive. We address her/his specific remarks below.

1. Can the authors show if CaM-binding mutants of TRPV5, such as TRPV5- Δ 696-729, TRPV5-W702A or TRPV5-R706E mutants have any difference in the ER retention compared to WT TRPV5?

We have repeated biotinylation experiments to assess the abundance of TRPV5 in the whole cell and on the plasma membrane using the suggested panel of TRPV5 mutants that are involved in CaM binding. While this is not a direct assay to investigate ER retention (as shown in Figure 4), this does show potential intracellular storage, which could be in the ER, or in intracellular (recycling) vesicles.

While the T709A mutation is able to bind CaM to a similar extent as WT TRPV5, TRPV5 W702A, R706E and C-terminal truncation mutant 698X all show impaired CaM binding (ref: doi: 10.1128/MCB.01319-10, doi: 10.1038/srep45489). We aimed to identify whether the regulatory effects of CaM on TRPV5 plasma membrane abundance is dependent on TRPV5-CaM interaction. The total TRPV5 protein abundance is not significantly different between WT TRPV5 and TRPV5 mutants W702A, R706E and T709A and C-terminal truncation mutant 698X (Figure 6A in revised manuscript). Compared to WT, TRPV5 mutants W702A, R706E and 698X have a higher abundance on the plasma membrane. (Due to the variation in effect size between the 3 independent experiments, only the abundance between WT and 698X is statistically significantly different in the post hoc analysis (Figure 6B in revised manuscript)). In line with the CaM siRNA experiments, this suggests that the lack of TRPV5-CaM interaction augments the abundance of TRPV5 on the plasma membrane. As the total TRPV5 protein expression is not different between cells transfected with WT or a CaM-binding mutant, this indicates that CaM retains TRPV5 in intracellular stores.

2. Did the authors try a Calmodulin Inhibitory peptide to test if that prevents the retention of TRPV5 in the ER?

We assessed the effect of membrane permeable CaM inhibitor W-7 in biotinylation experiments. W-7 hydrochloride (MedChemExpress, NJ, USA) was dissolved at 100 mM in DMSO and used in a final concentration of 50 μ M. Increased W-7 concentrations resulted in cell detachment.

HEK293 cells were transfected with eGFP-hTRPV5 and were treated with W-7 or DMSO for 10 minutes before performing a biotinylation assay to determine TRPV5 abundance on the plasma membrane.

Figure. Effect of W7 on TRPV5 expression.

The assay was performed 9 times (triplo's in 3 independent experiments) and the fraction of TRPV5 on the membrane (corrected for total TRPV5 expression) is plotted on the right.

Due to the inconsistency between the assay repeats, we are not confident that the W-7 inhibitor works for this specific assay in our hands. We have therefore decided not to include these results in the revised manuscript.

3. In the introduction section, it may be good to briefly mention that CaM is known to regulate the trafficking of several ion channels such as CaV1.2, KCNQ2, SK4/IK1 and others.

We thank the reviewer for this suggestion to make the introduction more comprehensive. A section on the role of CaM in the trafficking of other relevant ion channels has been now included in the introduction.

4. It is not the best idea to not talk anything about the CaM in section 1 and 2, given its co-expression in the presented data. Consider either reminding the rationale for using CaM in these experiments or presenting the data with cells that don't overly express CaM. Ideal would be to kick off with the localization of TRPV5 without the over-expression of CaM.

We appreciate the reviewer suggestions to clarify these sections of our study. We have now added the rationale for co-expressing CaM in these experiments in the manuscript text.

5. In Figure 6, it would be valuable for the authors to elaborate on how they discern whether the heightened intracellular Ca²⁺ in siCaM cells arises from an augmented surface abundance of CaM or from a potential failure in CaM-mediated inactivation of TRPV5. Addressing this aspect would enhance the depth of the discussion.

We agree with the reviewer that the Fura-2 experiments cannot discern whether the heightened intracellular Ca²⁺ in CaM siRNA-treated cells arise from an augmented surface abundance of TRPV5 or from a potential failure in CaM-mediated inactivation of TRPV5. While this is indicated in the manuscript text, we have now included a section in the discussion advocating for the inclusion of membrane abundance experiments in further studies into TRPV5 regulation.

Dear Professor Hoenderop,

Re: JP-RP-2024-286182R1 "Calmodulin regulates TRPV5 intracellular trafficking and plasma membrane abundance" by Malou Zuidsherwoude, Teodora Grigore, Brenda van de Langenberg, Guusje Witte, Jenny van der Wijst, and Joost G Hoenderop

We are pleased to tell you that your paper has been accepted for publication in The Journal of Physiology.

Yours sincerely,

Peying Fong
Senior Editor
The Journal of Physiology

If you would like to receive our 'Research Roundup', a monthly newsletter highlighting the cutting-edge research published in The Physiological Society's family of journals (The Journal of Physiology, Experimental Physiology, Physiological Reports, The Journal of Nutritional Physiology and The Journal of Precision Medicine: Health and Disease), please click this link, fill in your name and email address and select 'Research Roundup':

<https://www.physoc.org/journals-and-media/membernews>

- You can help your research get the attention it deserves! Check out Wiley's free Promotion Guide for best-practice recommendations for promoting your work at: www.wileyauthors.com/eo/guide. You can learn more about Wiley Editing Services which offers professional video, design, and writing services to create shareable video abstracts, infographics, conference posters, lay summaries, and research news stories for your research at: www.wileyauthors.com/eo/promotion.

The Corresponding Author will receive an email from Wiley with details on how to register or log-in to Wiley Authors Services where you will be able to place an order

Reviewing Editor's comments:

Thank you for thoroughly addressing the Referees' comments and suggestions.

Senior Editor's comments:

Both Expert Referees and the Reviewing Editor agree that the present version of your manuscript bears potential for high impact on a still-growing and important field. Thank you for submitting your work to The Journal of Physiology. On behalf of the Reviewing Editor and the Referees, congratulations!

Referee #1:

The authors have responded carefully and satisfactorily to all of the comments that were made.

Referee #2:

The revised manuscript JP-RP-2024-286182R1 titled "Calmodulin regulates TRPV5 intracellular trafficking and plasma membrane abundance," submitted by Zuidsherwoude et al. is of acceptable quality as the authors have addressed the

comments and concerns raised. I, therefore, recommend the manuscript for acceptance.

END OF COMMENTS

1st Confidential Review

15-Oct-2024